

# Can rifts alter ocean dynamics beneath ice shelves?

Mattia Poinelli[1,2,3], Michael Schodlok[1], Eric Larour[1], Miren Vizcaino[3], and Riccardo Riva[3]

[1]Jet Propulsion Laboratory, California Institute of Technology, Pasadena, CA
[2]Department of Earth System Science, University of California, Irvine, Irvine CA
[3]Department of Geoscience and Remote Sensing, Delft University of Technology, Netherlands

**Correspondence:** Mattia Poinelli (mattia.poinelli@jpl.nasa.gov)

**Abstract.** Land ice discharge from the Antarctic continent into the ocean is restrained by ice shelves, floating extensions of grounded ice that buttress the glacier outflow. The ongoing thinning of these ice shelves - largely due to enhanced melting at their base in response to global warming - is known to accelerate the release of glacier meltwater into the world oceans, augmenting global sea level. Mechanisms of ocean heat intrusion under the ice base are therefore crucial to project the future

of Antarctic ice shelves. Furthermore, ice shelves are weakened by the presence of km-wide full-thickness ice rifts, which are observed all around Antarctica. However, their impact on ocean circulation around and below ice shelves has been largely unexplored as ocean models are commonly characterized by resolutions that are too coarse to resolve their presence. Here, we apply the Massachusetts Institute of Technology general circulation model at high resolution to investigate the sensitivity of sub-shelf ocean dynamics and ice shelf melting to the presence of a km-wide rift in proximity of the ice front. We find that (a)

the rift curtails water and heat intrusion beneath the ice shelf base and (b) basal melting of a rifted ice shelf is on average 20 % lower than for an intact ice shelf under identical forcing. We therefore posit that rifts and their impact in the sub-shelf dynamics are important to consider in order to accurately reproduce and project pathways of heat intrusion into the ice shelf cavity.

## 1 Introduction

The Antarctic Ice Sheet (AIS) is losing mass at an accelerated pace (Paolo et al., 2015). Between 2001 and 2017, its ablation rate was approximately 134 Gigatons of ice every year (Gt yr$^{-1}$), with an estimated acceleration of 94 Gt yr$^{-1}$ each decade since 1979 (Rignot et al., 2019). While the current ice loss from the AIS contributes to 0.43 mm of global sea level rise every year (Poertner et al., 2019), the continent stores an amount of freshwater that - if completely released in the world oceans - would rise the global sea level by 58 m (Morlighem et al., 2020). Recent projections indicate an increase in the future

commitment of the AIS to global sea level, by up to 2.8 m within the next 300 years (Poertner et al., 2019) in response to past and ongoing global warming.

     The discharge of glacier ice from the Antarctic continent into the ocean is restrained by floating extensions of the AIS along most of its perimeter. These extensions are called ice shelves and form when the seaward downstream of continental





ice detaches from the underlying bedrock and becomes afloat at a location called grounding line. Ice shelves are critical in the

stability of the AIS as they act as a "cork in the bottle", by buttressing upstream grounded ice from further sliding into the ocean (Dupont and Alley, 2005). Therefore, loss of floating ice shelves - despite not directly adding up to sea level - leads to accelerated mass discharge from outlet glaciers into the ocean (Rignot et al., 2004; Scambos et al., 2004), rising the global sea level. So understanding the evolution of ice shelves is crucial to reconstruct Antarctic mass loss.

The strongest ablation process of Antarctic ice shelves occurs at their base (Rignot et al., 2013), where ice is flushed by ocean

water and melts from underneath. As the southern ocean is currently warming (Schmidtko et al., 2014) in response to climate change, the AIS mass loss is accelerating (Paolo et al., 2015). Observations show that sustained melting at the grounding line - when warm water has direct access (e.g. Nakayama et al., 2018, 2019) - causes it to migrate landward (e.g. Rignot and Jacobs, 2002). This process leads to further ungrounding of continental ice and accelerates freshwater discharge from outlet glaciers into the ocean (e.g Thomas et al., 2004). Albeit critical, pathways of ocean heat toward the grounding line remain elusive as

direct measurements in these locations are challenging to obtain.

Ice shelves are often categorized as "cold water" or "warm water", depending on the water mass that dominates the continental shelf and the melting processes that drive basal ablation (e.g. Dinniman et al., 2016). In proximity of cold water ice shelves, a water mass that is commonly found is High Salinity Shelf Water. It is formed by brine rejection during winter sea ice formation and can downwell near the coast due to its increased density. This hypersaline water can intrude in deep sec-

tions of the ice cavity (Nicholls et al., 2009) and - despite being sourced at the surface freezing temperature (-1.9$^o$C) - melts the ice bottom because the freezing point of seawater ($T_\mathrm{fr}$) decreases with increasing pressure. These processes are observed under large ice shelves in the Weddell and Ross sea, where sea ice production is abundant. In contrast, small warm water ice shelves in the Bellingshausen and Amundsen sea are directly flooded by warm Circumpolar Deep Water (1$^o$C) which spills on the continental shelf, often with direct access to the grounding line (e.g. Nakayama et al., 2018, 2019). These ice shelves are

rapidly thinning as a consequence (e.g. Shepherd et al., 2004).

Iceberg calving represents the second largest ablation process of Antarctic ice shelves (Rignot et al., 2013). The seaward extension of ice shelves is often curtailed by full-thickness ice rifts - usually propagating in a direction transverse to the ice flow - that episodically separates ("calves") large tabular icebergs. While ice rift width is usually less than 10 km, their length can extend up to 300 km. If rifting causes the ice front to recede inland of the so-called "compressive arch", the buttressing

force of the ice shelf may be compromised (Fuerst et al., 2016) and its disintegration may become irreversible (Doake et al., 1998).

Evidence also shows that rifts are often capped with ice mélange - a conundrum of accreted ice, windblown snow and ice shelf fragments - whose thickness is one to two orders or magnitude larger than regular sea ice (Orheim et al., 1990; Breyer and Fricker, 2022) and possibly holds rift flanks together (Rignot and MacAyeal, 1998; Larour et al., 2021). Beside sea ice

accretion from atmosphere cooling, Khazendar and Jenkins (2003) showed that ice rifts can be filled with marine ice, growing from ocean freezing. According to this process, buoyant plumes - formed after light meltwater is released in lower parts of the rift sidewalls - may freeze after relative cold seawater drops below $T_\mathrm{fr}$ in shallower sections of the rift, as the freezing temperature of seawater decreases with depressurization ("ice-pump" mechanism, Lewis and Perkin, 1983). This theory could





explain (a) the presence of water below $T_{fr}$ observed in rifts (Orheim et al., 1990; Lawrence et al., 2020) and (b) the possible
downwelling of dense water to compensate the upwelling at the sidewalls (firstly hypothesized by Potter and Paren, 1985).
Further numerical simulations of Jordan et al. (2014) confirmed these results by arguing that ice deposition in upper parts of
the rift is a key driver in the overturning circulation within ice fractures.

Nevertheless, previous studies on rift-ocean interactions are limited to a 2-D rift environment and neglect the impact of
rifts and water formed within in the dynamics of deeper sections of the ice shelf cavity, despite the known importance of
accurately represent the cavity geometry (Goldberg et al., 2012; Schodlok et al., 2012). Moreover, regional ocean models
of Antarctic ice shelves usually employ resolutions that are too coarse to capture km-wide rifts, hence their interaction with
the sub-shelf dynamics is often neglected. Mixing processes under rifted ice shelves and pathways of heat intrusion remain
therefore particularly unresolved.

In order to determine whether ice rifts actually impact the ocean dynamics beneath ice shelves, in this paper we assess (a)
the sensitivity of sub-shelf ocean circulation and basal melt distribution to the presence of a km-wide ice-capped rift near the
ice shelf front and (b) the impact of such a rift on the intrusion of off-shelf water. To this end, we run a 3-D ocean model with
idealized bathymetry and ice shelf geometry, at a resolution (250 m) that can capture a km-wide rift. To further investigate the
impact, we employ a suite of sensitivity simulations for a range of rift widths, thermal ocean forcing and magnitude/direction
of off-shelf ocean currents.

## 75  2  Methodology

### 2.1  Model set-up

To perform our sensitivity study, we use the Massachusetts Institute of Technology general circulation model (MITgcm).
Ocean dynamics is governed by the hydrostatic 3-D Navier-Stokes equations under the Bousinessq approximation. A non-
linear second-order flux-limiter advection scheme is applied. The equation of state follows the modified UNESCO formula
(Jackett and McDougall, 1995). Eddy diffusivities of temperature and salinity are held constant to $5.44 \times 10^{-7}$ m$^2$s in the
vertical dimension, while we use a lateral biharmonic Leith eddy viscosity factor of 2. Vertical ocean mixing is governed by
shear instability and internal wave activity, and approximated in MITgcm with the non-local K-Profile parameterization (KPP,
Large et al., 1994). Ice-ocean processes at horizontal and vertical interfaces are represented by the three-equation model of
Hellmer and Olbers (1989). Equations and adopted approximations are sketched in Appendix A.

We employ an idealized ice shelf-ocean model configuration that relies on a Cartesian coordinate system (100 km x 200 km
x 600 m, Fig. 1a-b) on a $f$-plane with Coriolis frequency set to $-1.4 \times 10^{-4}$ rad s$^{-1}$. Horizontal and vertical grid spacings are
set to 250 m and 10 m, which constrain the time-step to 30 s in order to ensure numerical convergence. The model relies on an
isotropic mesh and uses a partial cell formulation (Adcroft et al., 1997), with a minimum open cell fraction of 30 %.

The ice shelf thickness at the ice front (located at km 85 east) is set to 200 m slightly increasing to 250 m from km 35 east
along the ice shelf slope towards the grounding line (km 0 east). The bathymetry consists of a prograde slope, deepening from
the grounding line depth of 250 m to a maximum depth of 600 m at the eastern boundary (km 100 east).





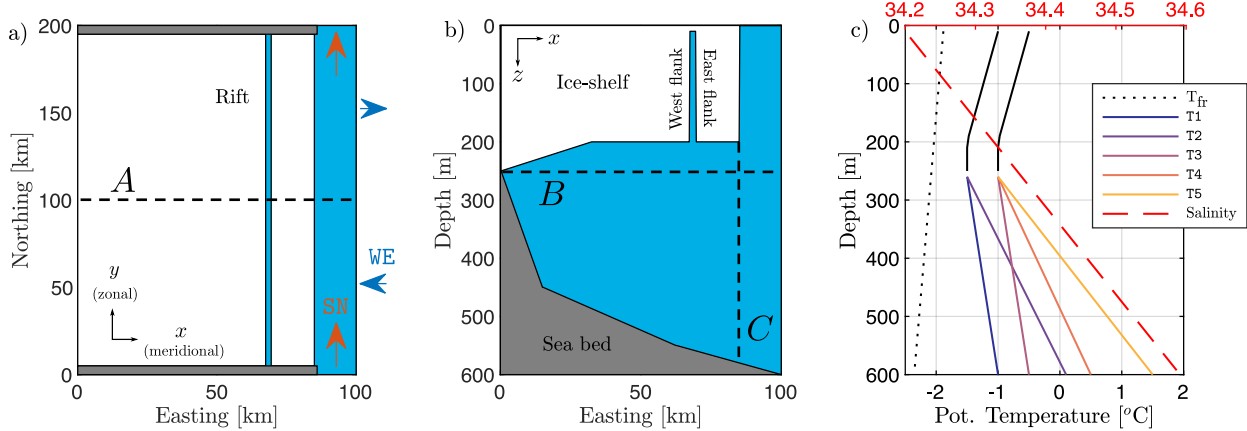

**Figure 1.** Model domain sketches and ocean profiles. Panel a shows a schematic view of an horizontal cross-section of the ice shelf (white), rift and ocean (blue), and seabed (grey) at 150 m depth. Red and blue arrows are directions of the alternating boundary conditions, which are termed here as south-to-north (SN) and west/eastward (WE) flow. In the simulations with a rift, the west flank of the rift is fixed at km 67.5 east. Beside runs with an intact shelf, the rift is introduced with a width of 1, 2 and 3 km and is filled with 15 m of ice shelf. Panel b shows a schematic view of a vertical cross-section of the model domain at km 100 north. Black dashed lines labeled $A$, $B$ and $C$ are horizontal and vertical cross-sections useful to visualize model results. Panel c shows potential temperature and salinity profiles as initial and boundary conditions of the sensitivity experiments. Solid lines are initial and boundary conditions of temperature for "cold water" and "warm water" regimes. Cold and warm profiles are characterized by two (T1, T2) and three (T3, T4, T5) bottom temperatures. Red dashed line is the salinity profile, constant in all experiments. Black dotted line is the freezing temperature of seawater ($T_{fr}$).

## 2.2 Simulations design

We performed a total of 120 simulations with combinations of rift width, ocean temperature, intensity and direction of ocean currents flowing in the open ocean (Table 1).

Our simulations include a rift as discontinuity in the ice shelf draft. Its west flank is located at a fixed distance (67.5 km) from the grounding line and rifts are introduced with a width of 1, 2 or 3 km. Furthermore, the rift is filled with 15 m of ice shelf, as a buffer layer between atmosphere and ocean, which approximates an "ice-capped" rift and does not aim to reproduce ice mélange properties. Despite ice mélange is often thought to be controlled by processes that are similar to those of seaice, evidence shows that the heterogeneous material filling km-wide rifts reaches thicknesses that are one to two order of magnitude

larger than sea ice (Orheim et al., 1990; Breyer and Fricker, 2022). Such a thick layer of ice can completely separate the liquid ocean from the atmosphere (Khazendar and Jenkins, 2003). By adopting the "ice-capped" approximation, we only focus our analysis on sub-shelf ocean processes introduced by the rift, while atmospheric and sea ice formation processes in the open ocean are neglected in this study. Freezing/melting processes are active on horizontal and vertical interfaces between ice and ocean: ice base, rift top, ice front and rift sidewalls.





Ocean circulation in the open ocean is prescribed as an inflow of varying magnitude (1, 5 and 10 cm s$^{-1}$) as Dirichlet boundary condition at the open ocean section of the south, north and east boundaries We perform two sets of experiments with alternating open boundaries (Fig. 1a). In the first - which we term here as south-to-north flow case (`SN`) - we "close" the eastern boundary (non-penetration condition) and prescribe an inflow at the southern boundary which is balanced by an outflow at the northern boundary to conserve mass. In the second - which we term here as west/eastward flow case (`WE`) - we close northern

and southern boundaries and, at the eastern bound, we impose an inflow between km 0 - 100 north which is balanced by an outflow between km 100 - 200 north. The `SN` scenario can be seen as an idealization of along-ice front ocean currents, while `WE` cases represent across-shelf currents.

We initialize and force the simulations with idealized temperature-salinity profiles, distinguishing between "warm" and "cold" regimes, which are prescribed as initial and boundary conditions (Fig. 1c). Each profile is characterized by a mixed

layer for the upper 250 m which linearly changes to (a) two different bottom temperatures in the cold regime, and (b) three different bottom temperatures in the warm regime (-0.5$^o$C in experiment `T3`, 0.5$^o$C in `T4` and 1.5$^o$C in `T5`). The salinity profile increases linearly from 34.2 psu at the surface to 34.6 psu at 600 m depth in all experiments. The chosen temperature profiles are representative of temperature ranges observed nearby ice shelves in Antarctica (Orsi and Whitworth, 2005). Warm water ice shelves in the Amundsen and Bellingshausen sea are represented by experiments under `T5` forcing. In contrast, cold scenarios

under `T1` are similar to real conditions of cold water ice shelves in the Weddell and Ross sea. The remaining scenarios (`T2-4`) can be seen as representative of "in-between" ice shelves, e.g. Totten Ice Shelf on the Sabrina Coast.

**Table 1.** Matrix of sensitivity experiments. Our study addresses the sensitivity of sub-shelf ocean dynamics to: four rift width configurations, two flow directions, three flow velocities and five temperature profiles, for a total of 120 experiments. In the following naming schema, R refers to rift width of 0 (intact shelf), 1, 2 or 3 km, `SN/WE1-10` refers to velocity at the open boundaries for south-to-north and west/eastward cases with intensity of 1, 5 or 10 cm s$^{-1}$ and `T` refers to temperature. Experiments highlighted in bold are chosen as representative cases and discussed in Sect. 3.2-3.4.

|  | T1 | T2 | T3 | T4 | T5 |
|---|---|---|---|---|---|
| Intact shelf | R0.SN/WE1-10.T1 | **R0.SN/WE1,5,10.T2** | R0.SN/WE1-10.T3 | R0.SN/WE1-10.T4 | R0.SN/WE1-10.T5 |
| 1 km rift | R1.SN/WE1-10.T1 | R1.SN/WE1-10.T2 | R1.SN/WE1-10.T3 | R1.SN/WE1-10.T4 | R1.SN/WE1-10.T5 |
| 2 km rift | R2.SN/WE1-10.T1 | **R2.SN/WE1,5,10.T2** | R2.SN/WE1-10.T3 | R2.SN/WE1-10.T4 | R2.SN/WE1-10.T5 |
| 3 km rift | R3.SN/WE1-10.T1 | R3.SN/WE1-10.T2 | R3.SN/WE1-10.T3 | R3.SN/WE1-10.T4 | R3.SN/WE1-10.T5 |

## 2.3    Formal analysis

All experiments are integrated for seven years. A quasi-steady state is reached in all simulations during the fifth year. To assess the impact of the rift in the sub-shelf dynamics, we focus on differences in modeled melt rate (equations in Appendix A1), heat



transport toward the ice shelf interior (equations in Appendix A2) and sub-shelf circulation between simulations with intact and rifted ice shelves using average values from the last year of integration, unless differently specified.

The discussion on the sub-shelf circulation and entrainment of water masses under the ice shelf is supported by the temporal evolution of virtual passive tracers. Tracers are initially released in the ice shelf cavity (`tr01`) and in the rift (`tr02`), with concentrations set at 1.0 at the beginning of the fifth year. Passive tracers do not affect the local density and advect with the

local water flow for two years, until the end of the simulations.

## 3  Results

Section 3.1 presents and discusses spatially-averaged melt rates and heat transport toward the interior of the ice shelf cavity across the 120 experiments, with emphasis on the sensitivity of basal melt to rift widths, thermal ocean forcing and magnitude/direction of off-shelf ocean currents.

Furthermore, we focus on the differences between intact and rifted simulations while sensitivity parameters are fixed to temperature profile `T2` and current velocity of 5 cm s$^{-1}$ in either `SN` or `WE` direction, unless differently specified. This sub-set is chosen as representative simulations. Section 3.2 shows the distribution of melt rate along the ice shelf base discussing differences between intact and rifted experiments. Moreover, Sect. 3.3 shows the distribution of melt and freeze rates along the rift surfaces. Finally, section 3.4 discusses the sub-shelf circulation by analysing barotropic streamfunction and the evolution

of passive tracers `tr01` and `tr02` under intact and rifted ice shelves. The temporal evolution of the vertical distribution of `tr01` across a vertical cross-section of the model domain (255 m, cross-section $B$ in Fig. 1b) is provided as video supplement.

### 3.1  Average melt rate across all simulations

In order to summarize the impact of the selected sensitivity parameters to ice shelf melting, we calculate spatially-averaged heat fluxes (in the form of melt rate) integrated along (a) ice shelf base and (b) rift surfaces (both rift top and sidewalls are

active ice-ocean interfaces) for the 120 simulations (Fig. 2). Intrusion of off-shelf water supplies the heat for melting the ice base, and our simulations show that heat transport toward the grounding line (calculated across the ice front, Appendix A2) increases with higher thermal forcing (Fig. 3), with a consequent increase in basal melt rate for higher ocean temperatures.

Furthermore, the intensity of basal melt depends on the direction and velocity of the applied boundary condition. In order to conserve potential vorticity, ocean flow tends to maintain its motion along lines of constant depth (isobaths). So water columns

are "stiff" in the vertical direction (Taylor columns) and tend to deflect along isobaths. In our case, isobaths are parallel to the ice front by model design. The abrupt gradient in the water column thickness that is introduced by the ice front therefore acts as a barrier that limits the flow entering in the cavity (e.g Grosfeld et al., 1997; Wåhlin et al., 2020).

Our `WE` experiments are a representation of across-shelf water currents that are forced perpendicularly to the ice front and are therefore associated with stronger on-shelf mass and heat intrusion as the forced flow across isobaths intensifies. As a

consequence, basal melt increases with stronger `WE` flow and large melt occurs near the ice front (further discussion in Sect. 3.2), regardless of thermal forcing and/or rift presence/width. On the other hand, `SN` simulations are representative of along-





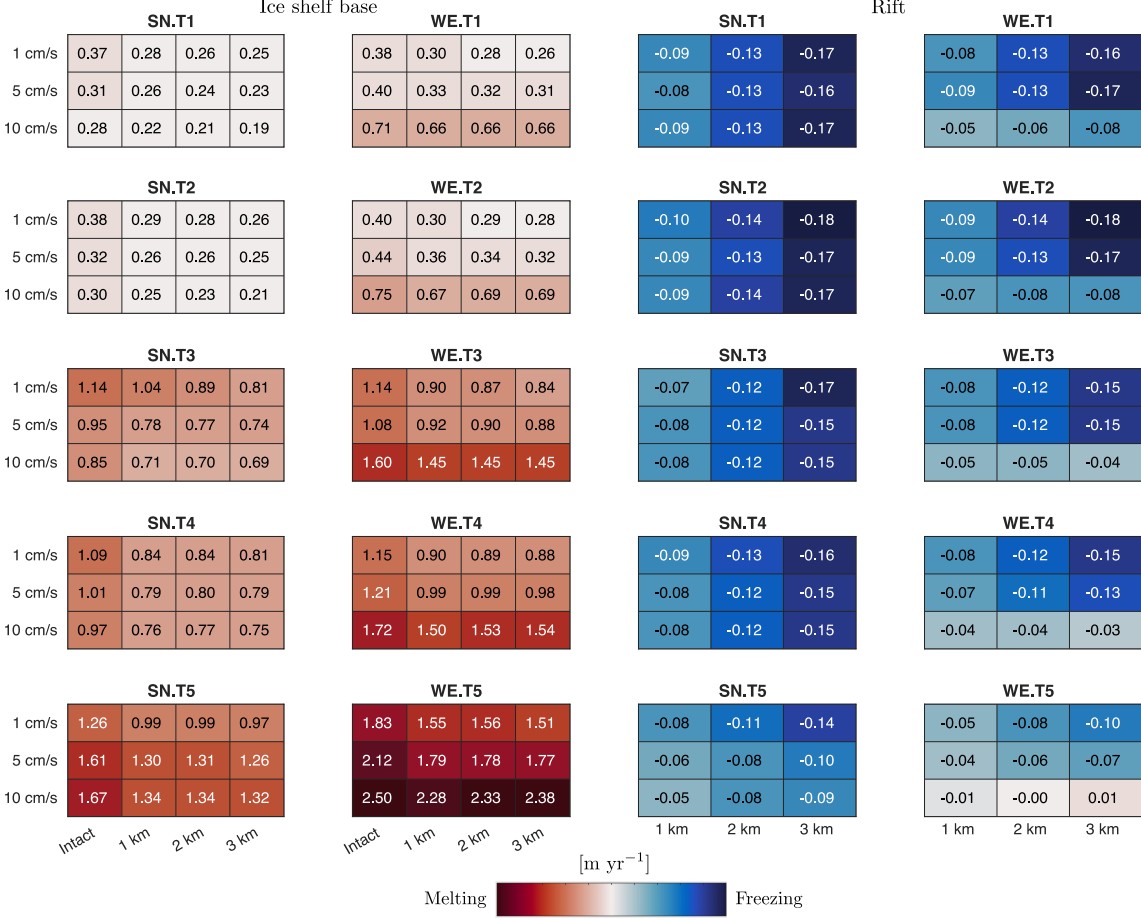

**Figure 2.** Heat-maps of average melt rate integrated along the ice shelf base (column one and two) and in the rift (column three and four) for the 120 sensitivity experiments. Values are calculated with respect to temperature profile (panel), flow velocity (row) and rift width (column).

front water currents. Stronger `SN` currents therefore imply an enhanced tendency of the flow to follow along-front isobaths and, in turn, to further isolate the ice shelf cavity from mass and heat intrusion. As a consequence, `SN` experiments show a much lower heat intrusion when compared to `WE` experiments, which also results in large melt differences. Furthermore, we find that intact `SN` experiments under thermal scenario `T1` to `T4` show a drop in melt rate with strengthening of the boundary flow. The only set of `SN` experiments that shares with `WE` the same trend between melt and boundary flow intensity are experiments under `T5` forcing.

The reason behind this discrepancy in basal melt with respect to boundary flow between `SN` experiments `T1-4` and `T5` resides in the higher heat transport that the latter boundary flow implies (heat transport in `T5` is twice as large as in `T4`, Fig. 3, while `T4` temperature is only 1$^o$C colder than `T5`). Extreme warm cases under thermal forcing `T5` result into a much higher



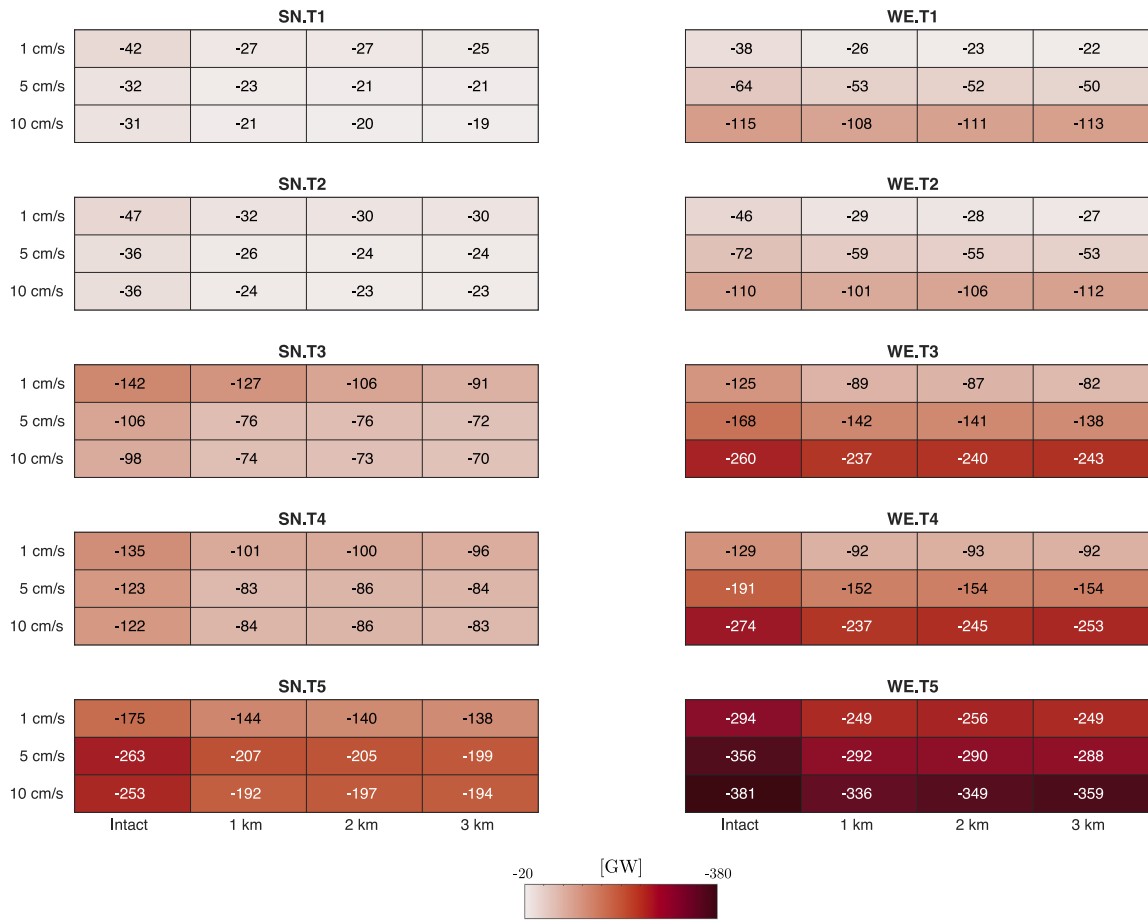

**Figure 3.** Heat-maps of total heat transport $HT$ (eq. A4) across the meridional cross-section passing throught the ice front (km 85 east, cross-section $C$ in Fig. 1b) for the 120 sensitivity experiments. The negative sign means that the heat transport is directed toward the ice cavity interior. Values are calculated with respect to temperature profile (panel), flow velocity (row) and rift width (column).

heat transport across the ice front in response to a stronger flow, with a consequent higher average melt rate. In this case, the largest contribution to the average melt rate increase is indeed concentrated near the ice front.

On average across all experiments, melting in rifted simulations is 20 % lower and the heat intrusion is 20 % weaker than what computed for intact shelves under identical forcing scenario. Our simulations also show that melt rate generally decreases with rift widening in all SN experiments and in WE experiments forced with a weak boundary flow (1 and 5 cm s$^{-1}$). The only exceptions are WE experiments under a 10 cm s$^{-1}$ boundary flow, where basal melt increases with rift widening. The differences in melting with respect to rift presence/width can be further investigated by evaluating heat fluxes in the rift.

The melt regime inside the rift is indeed very different from the ice shelf base, as we find that this environment is dominated by freezing in almost all experiments. The only exceptions are WE experiments at 10 cm s$^{-1}$ under thermal forcing T5, where



freezing in a 3 km rift is stopped. Following the three-equation model (Appendix A1), melting at the ice base freshens the ambient water and increases its buoyancy while freezing in the rift leads to water salinification and loss of buoyancy. These processes can also be quantified as a (negative) salt flux at the rift base (Fig. B1), whose intensity trends match' the dependency of freezing rate with respect to sensitivity parameters. Freeze distribution along the rift surfaces is further investigated in section 3.3.

Freeze rate in the rift shows the same dependency to thermal forcing as the basal melt rate discussed above. As the ocean temperature increases, freezing processes in the rift are inhibited and freeze rate decreases. However, differently than in the case of basal melt, we find that strengthening the boundary currents returns a negligible impact on the freeze rate in most experiments. The only exceptions are SN experiments under thermal forcing T5 and WE experiments at 5 and 10 cm s$^{-1}$, where the stronger currents leads to progressively lower freeze rate.

Finally, we find that freezing in the rift and the associated salt flux at the rift base increases with rift widening in almost all experiment with the only exception of WE experiments under a 10 cm s$^{-1}$ boundary flow. The anomaly of this last case can be explained with the analysis of melt/freeze pattern in the rift and its dependency to rift width (Sect. 3.3).

### 3.2   Melt pattern at the ice shelf base

In this section, we discuss the basal melt distribution along the ice shelf base and the calculated anomalies between intact and
rifted shelves.

   For simulations with an intact shelf, it is possible to distinguish three meridional regions (Fig. 4a,e). These are characterized by distinct melting regimes, which can be quantified with the spatially-averaged melt rate over the chosen regions (Fig. 5). We term these regions as grounding line (km 0 - 32.50 east), center (km 32.50 - 70.75 east) and front (km 70.75 - 85 east). The western-most region of intact experiments, adjacent to the grounding line, is dominated by similar average melt between SN
and WE cases. These are also the deepest sections of the ice shelf (255 m), and we show in Sect. 3.3 that fresh meltwater that is formed here tends to buoyantly rise along the ice-shelf slope. The shallower center region (200 m) shows large patches of water freezing, as water drops below the freezing temperature since this increases with lower pressure. On average, this region melts at negligible rate in both SN and WE intact experiments. Finally, the front region - directly in contact with water transport in the open ocean - returns melting rates that are about twice as large in the WE case with respect to the SN case.

The introduction of the rift results in a redistribution of heat along the entire ice shelf base, which can be quantified in each meridional region identified for intact cases (Fig. 5). Some of the impact is found in the central region, where the rift is actually located. In the intact case, this region is dominated by slow melting. Freezing processes that occur in the rift drive here a reduction in melt rate, which is progressively reduced by increasing the rift width. As a consequence, central regions of rifted cases show a generalized freezing. We indeed find that average freezing in the rift increases with larger rifts, with the
only exception of WE experiments under 10 cm s$^{-1}$. The general behavior is due to progressively larger area that is brought in contact with cold water dropping below the freezing point, which increases with wider rifts.

   The presence of the rift also leads to faster melting in the frontal region, where the model simulates an average increase in melt rate of 2 % in the SN case and of 12 % in the case of WE. Finally, our model results show that the region adjacent to



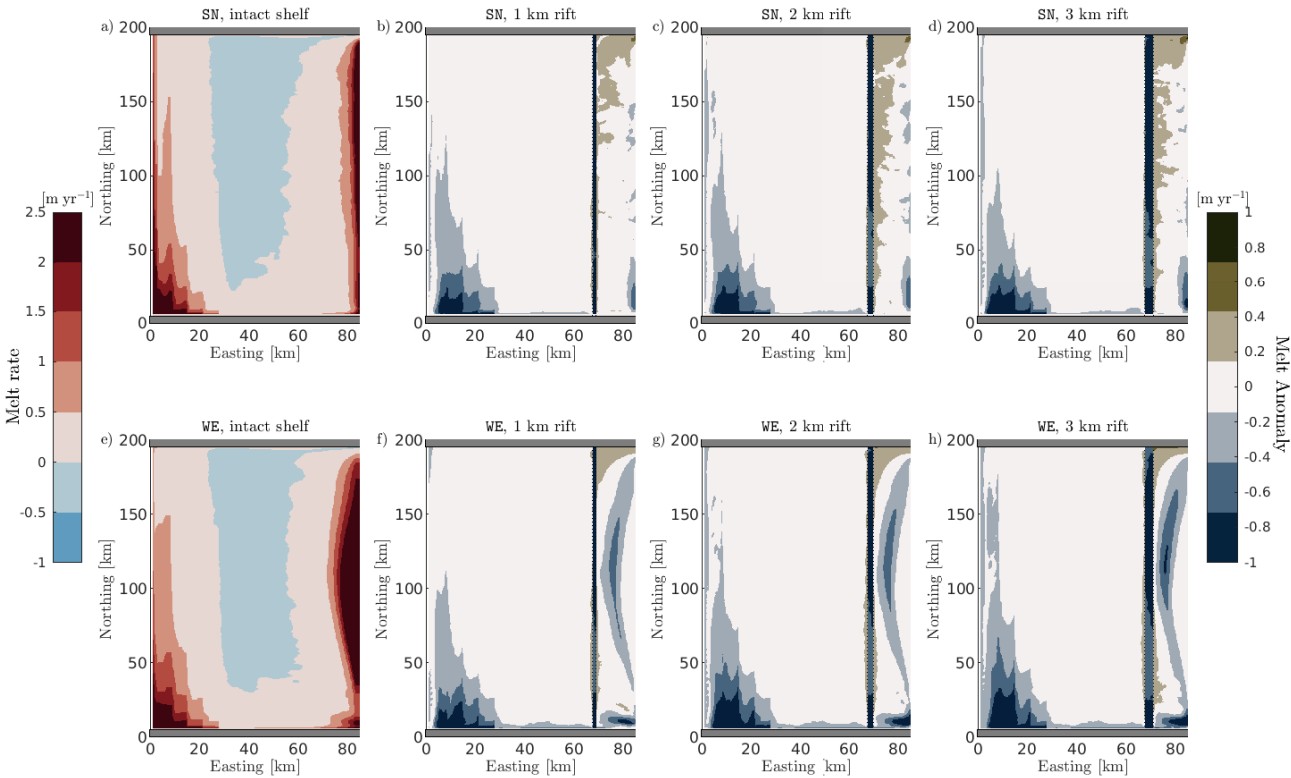

**Figure 4.** Comparison of horizontal distribution of ice shelf basal melt rates for different rift widths. Upper and lower panels correspond to SN and WE boundary currents of 5 cm s$^{-1}$, respectively. The cold temperature profile T2 is used for all simulations. The first column shows the horizontal distribution of melt rate of intact experiments (a,e). Panels b-d and f-h show the horizontal distribution of melt anomaly, which is calculated as the difference between melt rate in the rifted shelf simulations with respect to intact cases under the same forcing conditions. Left colorbar refers to melt rate, while right colorbar refers to melt anomaly.

the grounding line in rifted experiments returns the largest anomaly in the average melt rate with respect to intact simulations.

When the rift is introduced, the grounding line region results in 30 % less melt rate than in the intact case under both SN and WE. WE simulations also show a net drop in melt rate in this region with rift widening, while in SN cases rift widening results in a negligible impact.

## 3.3    Melt pattern in the rift

It has been already discussed how the majority of experiments show that, on average, the rift environment is dominated by

freezing (Fig. 2). In this Sect., we address how freezing patterns are distributed along the rift boundaries. The following



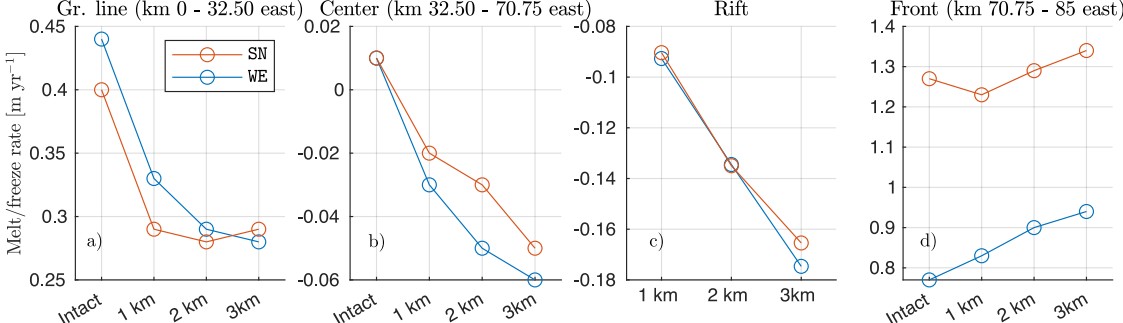

**Figure 5.** Spatially-averaged melt/freeze rate as a function of rift width for the three regions defined as grounding line, center and front. Values are referring to `SN` and `WE` experiments at 5 cm s$^{-1}$ while the temperature profile is fixed to `T2`. Positive means melting and negative means freezing.

analysis refers to `SN` and `WE` experiments with a 2 km wide rift and boundary flow at 5 cm s$^{-1}$ and 10 cm s$^{-1}$ under thermal forcing `T2`.

Along the rift sidewalls, our simulations show that melting occurs at their deeper levels and freezing at the top (Fig. 6). This bimodal pattern along the rift flanks is a result of the ice-pump mechanism active within the rift (Khazendar and Jenkins, 2003),

as the freezing point of seawater increases with decreasing pressure. Melting in deeper sections of the rift upwells cold and fresh meltwater along the west flank (Fig. 6), following an overturning pattern that is similar to the "melt-driven" experiments of Jordan et al. (2014). Meltwater reaches the freezing point at a depth of around 120 m and water re-freeze above this level and at the rift top. Freezing in shallow portions of the rift leads to salinification of the ambient water and a consequent loss of buoyancy. The upwell of meltwater along the west flanks is generally balanced by sinking of denser water along the east flank.

We term the water mass that interacts with the rift as Rift Water. This water is tracked by passive tracer `tr02`, which is released in the rift environment at the beginning of the simulations. The downward transport of Rift Water can also be quantified as a (negative) salt flux through the rift base (Fig. B1).

While the `SN` flow increases from 5 to 10 cm s$^{-1}$, the impact on the rift freeze is negligible (Fig. 6a-c to g-i). In contrast, strengthening the `WE` flow to 10 cm s$^{-1}$ (Fig. 6d-f to j-l) has a major impact in the heat budget in the rift, as stronger off-shelf

water is forced to penetrate deeper in western-most regions of the ice cavity. This water can therefore intrude within the rift environment along the east flank (6h), increases melt rate of deeper level of the rift flanks and inhibits freezing processes in shallower layers (Fig. 6j-l).

This behavior explains why `WE` experiments at 10 cm s$^{-1}$ result in a progressive decrease in freezing rate with rift widening. Forcing the model with a 10 cm s$^{-1}$ flow flushing the cavity perpendicularly to the ice front can reach inland-most section of

the ice cavity, when compared to weaker forcing and/or the `SN` set of experiments. As the east flank of the rift gets progressively closer to the ice front as the rift widens, off-cavity water eventually intrudes in progressively larger rifts (with a consequent switch in the sign of the salt flux across the rift base, Fig. B1) and inhibits freezing in within. This behavior also explains that average basal melting increases with rift widening in the case of experiments `WE` under a strong 10 cm s$^{-1}$ boundary flow, until





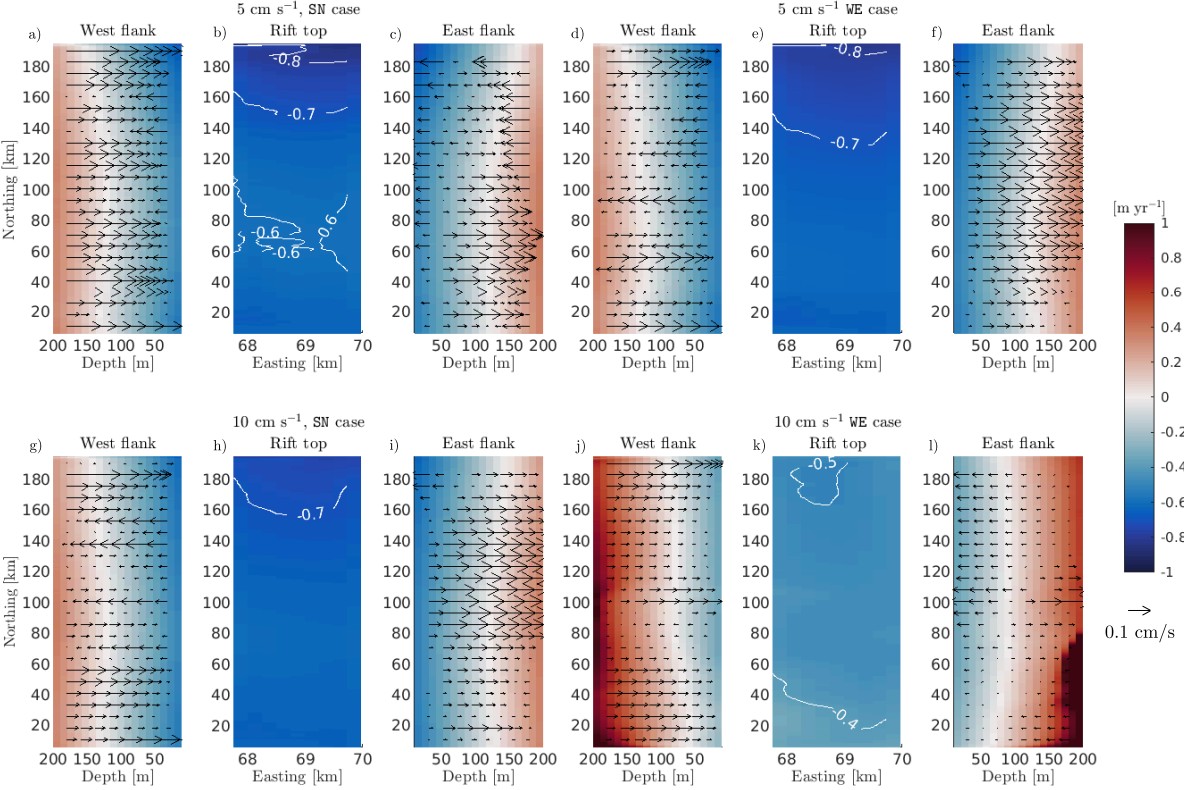

**Figure 6.** Comparison of melt rate distribution (both color-map and contour lines) along rift flanks and top of a 2 km wide rift under 5 and 10 cm s$^{-1}$ boundary forcing. The cold temperature profile T2 is used for all simulations. West flank is closer to the grounding line while the east flank is closer to the ice front (see Fig. 1b). Panels a-f correspond to SN and WE boundary currents of 5 cm s$^{-1}$, while panels g-n correspond to boundary currents of 10 cm s$^{-1}$. Positive means melting and negative freezing. Arrows are vertical velocity along the rift flanks.

the extreme case of WE under thermal forcing T5 with a 3 km rift shows that the rift is dominated by slow melting (Fig. 2).
On the other hand, the rest of the experiments (majority) show the opposite trend, where the larger the rift the wider the area subjected to freezing, hence the faster the average freezing rate.

## 3.4 Sub-shelf circulation

In this Sect., we address how sub-shelf circulation and ocean stratification are affected by the rift presence using modeled barotropic and overturning streamfunction together with the concentration of passive tracers tr01 and tr02 under the ice 245 shelf base.





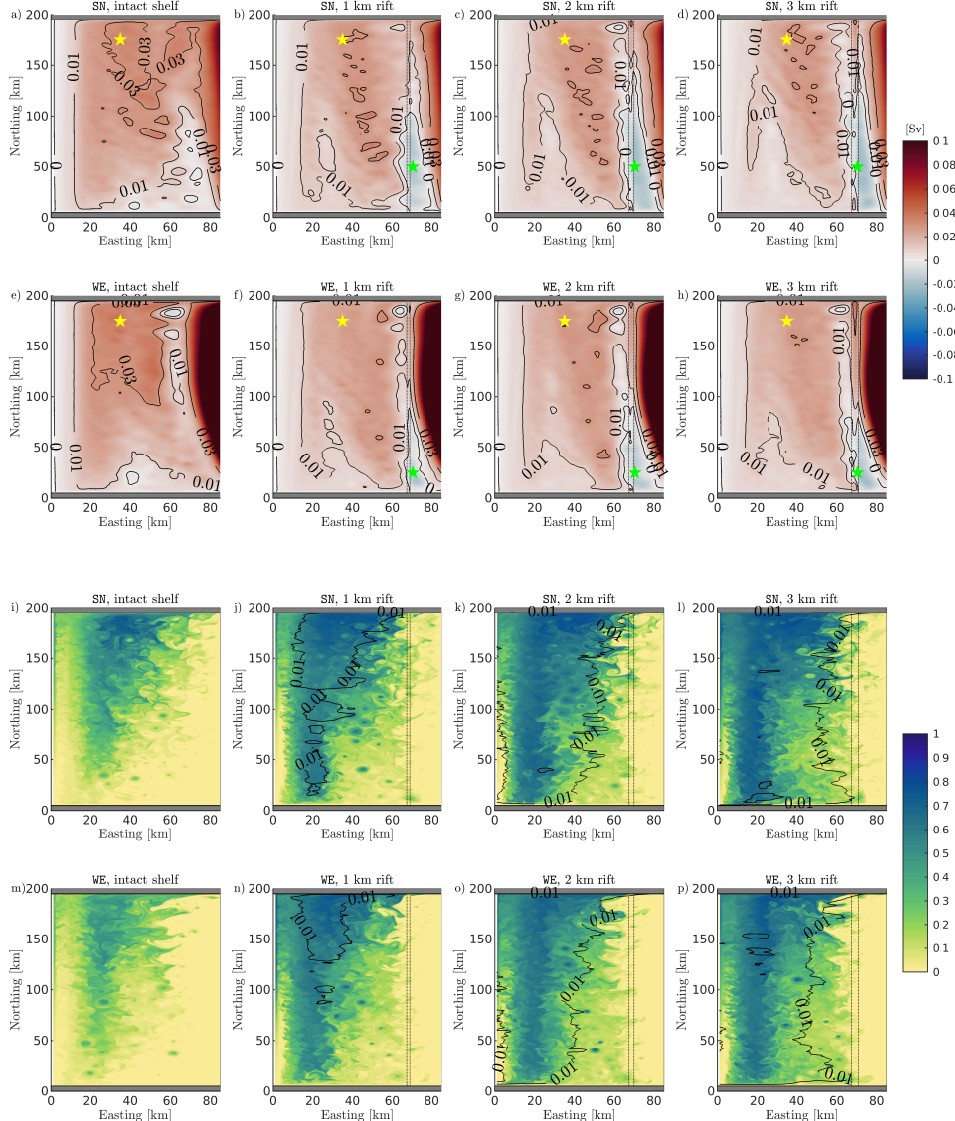

**Figure 7.** Comparison of barotropic streamfunction and horizontal concentration of passive tracers for experiments with different rift widths. First and third row (second and fourth row) correspond to `SN` (`WE`) boundary currents of 5 cm s$^{-1}$, respectively. The cold temperature profile `T2` is used for all simulations. Dashed line is the rift location. Panels a-h compare the horizontal distribution of barotropic streamfunction (both color-map and contour lines). Positive means clockwise rotation and negative means counter-clockwise rotation. Locations marked with yellow and green stars are used to calculate intensity and center of the two gyres. Panels i-p compare snapshots of the horizontal concentration of passive tracers `tr01` (background color-map) and `tr02` (black contour lines) at the grounding line level (255 m, cross-section $B$ in Fig. 1b) after 2 years from the initial release.



### 3.4.1 Horizontal circulation

Our simulations show that horizontal ocean dynamics under intact ice shelves is governed by the intrusion of off-shelf water, which is partially blocked by the ice front. The ice shelf front indeed represents a sharp discontinuity in the water column thickness that acts as if the topographic barrier extended along the entire water column (Taylor-Proudman theorem). This limits the amount of off-shelf water that entrains the cavity, which tends to flow along isobaths (parallel to the ice front by model design) in order to conserve potential vorticity.

The simulated water transport in the open ocean (not shown here) is almost 6 times stronger in the `WE` experiments when compared to the `SN` case, despite their prescribed velocity at the boundary is the same. As the former boundary condition is applied to the east boundary, the flux of water entering in the model domain is much larger than the latter. As a further consequence, water transport flowing in the open ocean penetrates much further in intact case forced with `WE` versus the case of `SN` (Fig. 7a,e). This also explains the larger melt rates in the proximity of the ice front for `WE` currents, where the ice is directly in contact with water transport in the open ocean (Fig 4a,e).

In both `SN` and `WE` experiments, inner regions of the ice shelf cavity are dominated by a cyclonic gyre, centered in the north section of the model domain (yellow star). This cyclonic system is driven by off-shelf water masses at -1.5$^o$C that entrains the cavity along the southern bound (Fig. 7i,m, Fig. C1a,e). This intrusion is also responsible to provide heat for melting the ice base (Fig. 3), while off-shelf water reaches and melts the western-most extension of the ice shelf cavity. In turn, melting along the grounding line region drives the formation of a buoyant plume of meltwater (at around -1.9$^o$C), which flows northward along the ice shelf slope (Fig. C1i,m) and returns toward the ice front along the northern edge, closing the circulation pattern.

In the case of a rifted ice shelf, the intrusion along the southern bound resembles intact experiments. Nevertheless, the intensity of mass and heat intrusion toward the grounding line is to a large extend curtailed. The inhibition is caused by the rift, that represents a second discontinuity in the water column thickness. In order to follow isobaths and conserve potential vorticity, the intrusive water flow along the southern edge is deflected along the rift (Fig. 7b-d and f-h), as shown by the fact that low values of `tr01` are confined seaward of the rift (Fig. 7j-l and n-p). On the other hand, `tr02` - which tracks Rift Water formed after freezing processes in the rift - entrains the cavity from the rift base and spreads along the ice shelf base replacing off-shelf water. As a result, areas of the ice shelf cavity eastward of the rift remains much cooler than in the intact case (Fig. C1a-h), and melt rate here is therefore reduced.

After the intrusion of off-shelf water is inhibited at the rift base, water transport westward of the rift is reduced as a consequence. The intensity of the cyclonic gyre progressively reduces by about 30 % in the case of a rifted shelf, with negligible difference between `SN` and `WE` experiments. Differently from the intact case, water transport from the open ocean along the southern boundary is separated from the cavity interior by the formation of second gyre - anticyclonic and less intense - which is centered under the rift (green star, Fig. 7b-d and f-h).

Rift width determines the strength of the individual gyres and to a lesser degree their dimension. While the cyclonic gyre decreases in strength as ocean transport in western-most region of the ice shelf cavity is weaker, the model calculates a pro-




gressive growth in the anticyclonic gyre size with rift widening for both `SN` and `WE` sets of simulations, as the cavity remains

colder (Fig. C1b-d and f-h) and melt rate reduces.

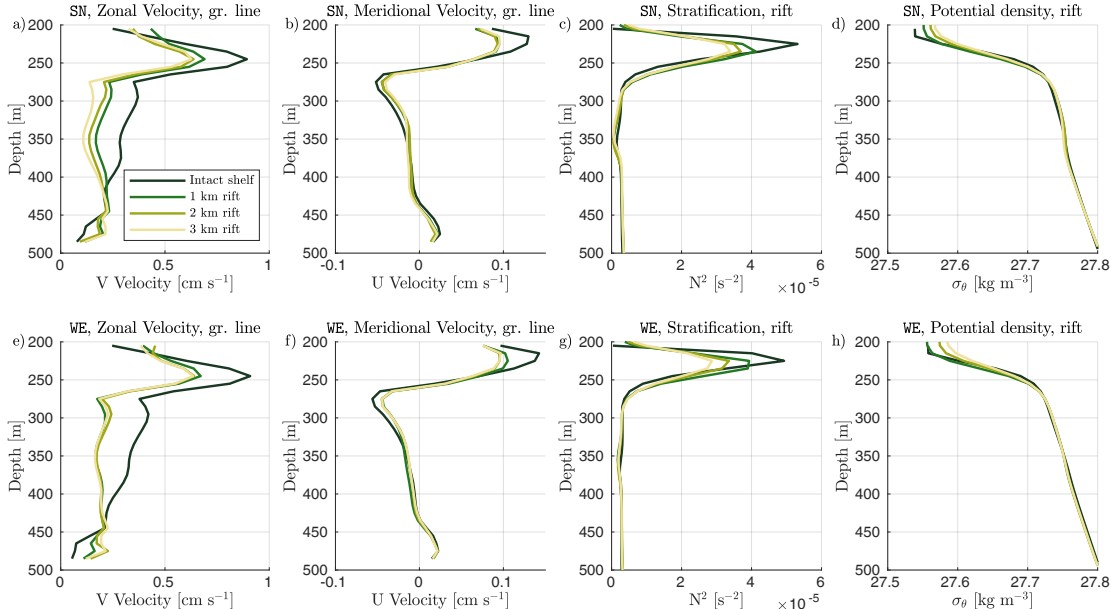

**Figure 8.** Comparison of vertical ocean profiles for experiments with different rift widths. Upper and lower panels correspond to `SN` and `WE` boundary currents of 5 cm s$^{-1}$, respectively. The cold temperature profile `T2` is used for all simulations. Vertical profiles of meridional and zonal velocities - as spatial average beneath the ice shelf slope (km 0 - 35 east, km 0 - 200 north) - are in panel a, b and e,f. Vertical profiles of buoyancy frequency squared $N^2$ and potential density anomaly ($\sigma_\theta = \rho_\theta - 1000$ kg m$^{-3}$, where $\rho_\theta$ is the local potential density) - as spatial average beneath the rift base (km 67.5 - 70.5 east, km 0 - 200 north) - are in panels c,d and g,h. Rift width is color-coded.

### 3.4.2   Overturning circulation

The calculated overturning streamfunction of intact experiments further shows that water transport under the ice base is to a large extent curtailed by the presence of the ice front (Fig. 9). However, a fraction of off-shelf water entrains the ice shelf cavity and the meridional on-shelf intrusion is strongest between 250 m and 300 m (Fig. 8b,f). This intrusion drives the formation of

a double overturning gyre system and delivers heat for melting the ice shelf in its deepest level, i.e. at the grounding line.

Simulations with intact shelf shows a clockwise cell which is centered in the shallow portion of the ice cavity, just beneath the ice base (yellow stars). Underneath the clockwise cell, our simulations show a further overturning transport of less intensity through a counter-clockwise gyre cell (green stars), as water returns toward the ice front along the seabed (below 450 m, profiles in Fig. 8b,f). Furthermore, the vertical ocean profile of intact experiments resembles a stable stratified ocean interior



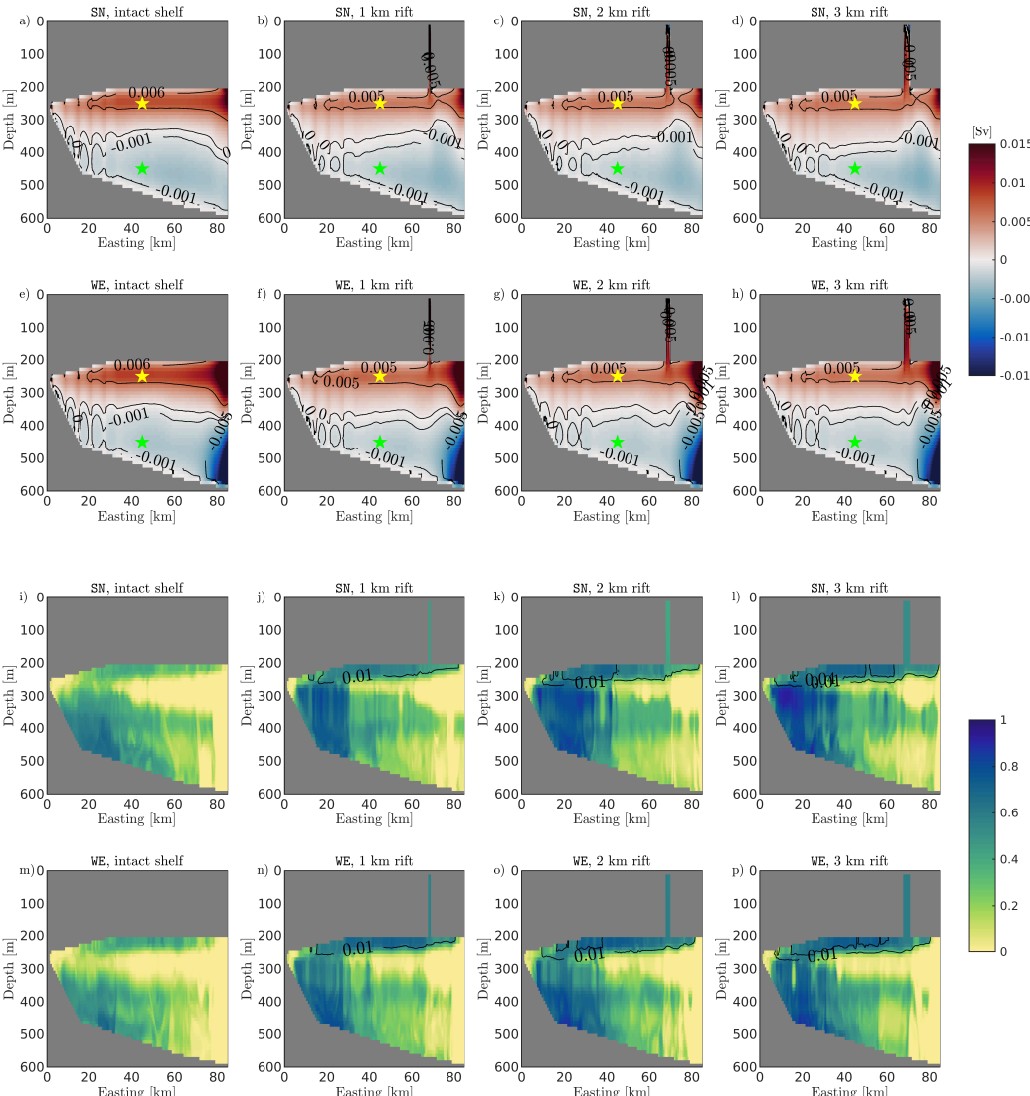

**Figure 9.** Comparison of overturning streamfunction and vertical concentration of passive tracers for experiments with different rift widths. First and third row (second and fourth row) correspond to SN (WE) boundary currents of 5 cm s$^{-1}$, respectively. The cold temperature profile T2 is used for all simulations. Panels a-h compare the vertical distribution of the overturning streamfunction (both color-map and contour lines). Positive means clockwise rotation and negative means counter-clockwise rotation. Locations marked with yellow and green stars are used to calculate intensity of the two overturning cells. Panels i-p compare snapshots of the vertical concentration of passive tracers tr01 (background color-map) and tr02 (black contour lines) at the zonal cross-section along km 50 north (cross-section $A$ in Fig. 1a)after 2 years from the initial release.





where fresh and light meltwater overlays denser water. The pycnocline - ocean layer where the density gradient is largest - is characterized by maximum stratification of $N^2 = 5 \times 10^{-5} \ s^{-2}$ calculated at a depth of 275 m (Fig. 8c-g).

In shallow layers of the model domain, water flows eastward along at the ice base, while melting at the grounding line is responsible for the increase in the buoyancy of the ambient water. The formed meltwater rises along the ice shelf slope (Fig. 8b,f, C2) and is replaced by off-shelf water flowing toward the grounding line at depths, closing the shallow overturning

circulation. Eventually, the upward flow of meltwater drops below the freezing point of seawater driving water freezing in the central region of the model domain (Fig. 4).

In order to conserve potential vorticity, the intrusive water flow along the southern edge is deflected along the rift - which introduces an abrupt change in the isobath lines - and the off-shelf intrusion between 250 and 350 m is curtailed in response. As a consequence, our rifted simulations show that the clockwise gyre under the ice base in both SN and WE sets of experiments

is 25% weaker than in the intact case while its continuity with transport in the open ocean is interrupted at the rift base.

Our model simulates that low values of tr01 are confined to seaward of the rift when this is present while water layers immediately beneath the ice shelf base are in contact with Rift Water (tr02). This water is formed after freezing in the ice rift and sinks due to decreased buoyancy. Sinking of Rift Water is also quantified by calculating the salt flux across the rift base (Fig. B1). The downwelling of Rift Water leads to spreading saltier and denser water in the shallower layers of the ice shelf

cavity (Fig. 8d-h), weakening the sub-shelf stratification by up to 40 % (Fig. 8c-g).

The impact of rift widening is particularly evident in the distribution of the two tracers. While the concentration of Rift Water along the ice base increases with larger rifts, the on-shelf penetration of water from the open ocean (low values of tr01) in western most areas of the ice cavity is progressively reduced with rift widening. While the density in the pycnocline increases, progressively stronger downwelling of Rift Water weakens the stratification under the ice shelf base, cools the ambient water

near the ice base and curtails basal melt.

## 4   Summary and discussion

Ice rifts and their impact in the circulation below and around ice shelves are generally neglected in ocean models, as employed resolutions are often too coarse to resolve km-wide fractures in the ice shelf. In this study, we use the MITgcm at high resolution to show that the presence of a prominent rift near the ice shelf front substantially alters the sub-shelf dynamics and the basal

melt distribution. A schematic diagram of our findings is sketched in Fig. 10.

Our idealized ocean model simulates that, under intact ice shelves, transport of off-cavity water is inhibited by the presence of the ice front. This acts as a discontinuity in the water column thickness and limits the amount of flow entraining the ice cavity. Indeed the water tend to flow along isobaths (in this case parallel to the ice front by model design) in order to conserve potential vorticity. A fraction of off-cavity water intrudes along the southern bound and provides the heat for melting the ice

base. In turn, light meltwater formed after melting at the grounding line - i.e. the deepest layers of the idealized ice shelf - buoyantly rises along the ice draft toward the ice front, eventually dropping below the freezing point of seawater, as this





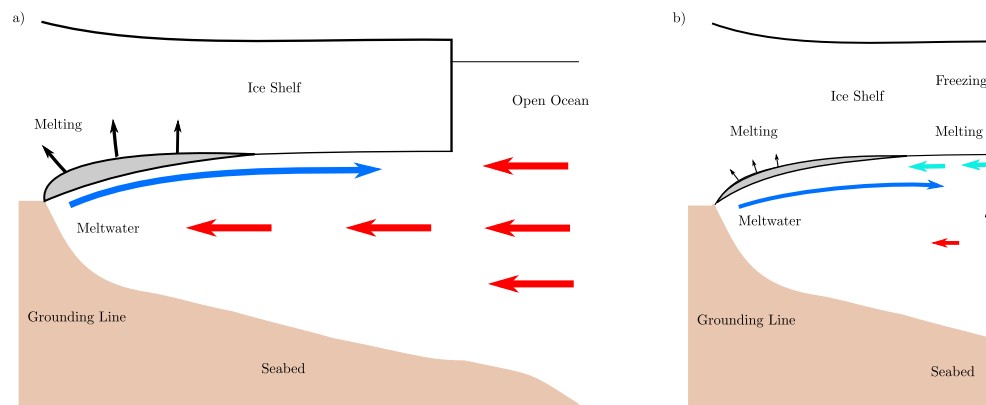

**Figure 10.** Schematic diagram of ocean dynamics underneath an intact ice shelf (a) and a rifted ice shelf (b), in our idealized model. The introduction of the rift interrupts the on-shelf intrusion of off-cavity water from the open ocean toward the grounding line, responsible of supplying heat for melting the ice base. As result, Rift Water (RW) floods the cavity from the rift base and basal melt is 20 % less than what computed in the intact case, while the largest melt reduction occurs in proximity of the grounding line.

decreases with depressurization. The sub-shelf circulation is closed as water returns toward the ice front along the northern edge (Fig. 7a,e).

We find that the introduction of an ice rift affects sub-shelf dynamics and distribution of basal melt by means of two
processes. Firstly, the off-shelf water intrusion - which reaches and melts the grounding line - is further inhibited by the introduction of the rift. This acts as a second discontinuity in the water column thickness (Fig. 9) after the ice front and leads off-shelf water to deviate along the rift (Fig. 7). Secondly, in the case of a rifted ice shelf, off-shelf water intruding in the ice base is replaced by the outflow of dense Rift Water from inside the rift. This water is formed after seawater reaches the freezing temperature higher in the rift (Fig. 6) causing it to salinify and sink. Modeled rift circulation is in good agreement
with past numerical efforts (e.g. Khazendar and Jenkins, 2003) and the simulated overturning circulation resembles the "melt-driven" experiments of Jordan et al. (2014). Moreover, our simulations show that the downward flow of Rift Water weakens the stratification of the water column by supplying denser water close to the ice base (Fig. 8). As a consequence, the ambient water under rifted ice shelves is much cooler than in intact experiments (Fig. C1a-h) and basal melt rate is curtailed in response (Fig. 4).

In perspective, the deflection of ocean currents along sharp discontinuities in the depth contour also occurs when the abrupt gradient is introduced at the bottom of the water column. For example, Menemenlis et al. (2007) showed that including a deep and narrow trench (equivalent of a rift but extruded in the seabed) passing through the middle of Gibraltar Strait from the Atlantic sea to the Mediterranean sea substantially reduces the ocean current transport across the strait. In those experiments, ocean currents are deflected along the trench and in part dissipated by outflowing from inside the trench, similarly to what
happens in our simulations after the intrusive flow enters the ice shelf cavity and interacts with the rift.



Across the tested range of parameters, we quantify that - in rifted simulations - the heat transport toward the grounding line is 20 % weaker and the melt rate is 20 % lower than in the case of an intact ice shelf under the same dynamic and thermodynamic forcing scenario (Fig. 2, 3). In particular, we estimate that melt rate in proximity of the grounding line (i.e. the deepest layers of the ice shelf) represents the largest anomaly with a decrease of 30 % when the rift is present. This result showcases the

potential of rifts formed near the ice front to impact the dynamics of the most vulnerable areas of ice shelves.

Our simulations indeed suggest that the impact of rifts in the sub-shelf circulation may have implications in the dynamics of the grounding line, whose migration is extremely sensitive to off-shelf water incursion (e.g. Seroussi et al., 2017). An accurate estimation of melt patterns at the grounding line is important as sustained melt in these vulnerable locations leads to further ungrounding of continental ice, which in turn accelerates ice discharge from outlet glaciers into the ocean (Rignot and Jacobs,

2002). Thanks to our high resolution model, we have shown how off-shelf water intrusion may be modulated by the presence of an ice rift near the ice front and by the formation of Rift Water from freezing processes within the rift. Rift Water may mix with water masses flowing toward the grounding line possibly re-distributing heat under the ice base. Finally, our simulations also show that the rift presence tends to modulate heat intrusion regardless of whether currents in the open ocean are directed across or along the ice front. As an precise representation of the ice shelf cavity geometry is essential to correctly reproduce

sub-shelf dynamics (Goldberg et al., 2012; Schodlok et al., 2012), we posit that high resolution ocean models should account for the presence of ice rifts in order to accurately reproduce heat intrusion under the ice shelf and the - yet largely unexplored - impact of rifts in the sub-shelf ocean dynamics.

Although our simulations introduce novel sub-shelf processes that have been neglected in past ice-ocean modelling efforts, our idealised model set-up cannot accurately quantify the effective importance of rifts in the sub-shelf dynamics of actual ice

shelves, as many critical variables are here ignored. In particular, sea ice processes near the ice front are key components in mixing processes under and near the ice shelf. For example, dense High Salinity Shelf Water is formed by brine rejection during winter sea ice formation. Beside being an important contributor to the Meridional Overturning Circulation and the Antarctic Bottom Water (e.g. van Caspel et al., 2015), High Salinity Shelf Water convects down near the coast due to its buoyancy. These cold and dense water masses flow in deeper sections of the ice cavity and perturb the water ambient at depth (e.g. Nicholls et al.,

2009). These mixing processes are controlled by atmospheric forcing, with implications that go beyond the scopes of our study. The inclusion of sea ice processes and their impact on water masses near the front of rifted ice shelf in future numerical efforts is therefore necessary to accurately understand ocean dynamics and heat intrusion in the ice shelf cavity. This is particularly relevant where winter sea ice production is abundant (e.g. large ice shelves in the Weddell sea). To this end, our work suggests that downwelling of dense Rift Water may imply a further layer of complexity in the mixing processes near the ice front, and

is therefore worth of further investigation.

Furthermore, rifts growth is episodic, with events that last from minutes to hours (e.g. Banwell et al., 2017), often followed by periods of inactivity of several months (Bassis et al., 2005; Walker et al., 2015). Rifts are often filled with ice mélange, whose rheological and mechanical properties are challenging to model (Vankova and Holland, 2017; Amundson et al., 2020). Although our simulations offer a highly simplified version of an ice rift - where atmospheric forcing and growth processes of

encased ice mélange are neglected and rift width is fixed - they suggest that changes in rift size will cause a direct modulation

of the amount of freezing between flanks (e.g. Fig. 2,4). Hence, mechanically-induced rift propagation - altering rift size, e.g due to ice heterogeneity (Borstad et al., 2017) - may directly influence thinning/thickening of encased ice mélange thus the cohesiveness of rift flanks, beside being triggered by variation of rift infill due to environmental warming (Holland et al., 2009; Larour et al., 2021).

## 5   Conclusions

Our study shows that resolving km-wide rifts in the geometrical description of the ice shelf cavity may imply a further layer of complexity in mixing processes below the ice base, with possible alteration of mass and heat intrusion. We estimate that these processes - so far largely unexplored - may alter heat intrusion and melt rate by as much as 20 % and are therefore worth of further investigation. On a larger scale, a better understanding of heat pathways toward the grounding line remains a

fundamental research avenue which would improve the estimation of ice shelf ablation and, consequently, the evolution of the Antarctic Ice Sheet in a warming climate.

*Code and data availability.* MITgcm is an open source numerical model, freely available on Github. All data-sets and model output comprises 27 TB of data and are stored on the Lou mass storage system at the NASA Advanced Super-computing facility at Ames Research Center. Data and model output used to produce figures and tables in this document will be available in Zenodo or Github, *pending publication*.

*Video supplement.* The supporting animation shows the two years evolution of passive tracer `tr01` distribution at the grounding line depth, 255 m, cross-section $B$ in Fig. 1b.

## Appendix A:  Thermodynamic equations

### A1   Three-equations model

Melting and freezing form an ice-ocean boundary layer which is implemented in MITgcm as a heat flux and a "virtual" salt
flux (i.e. a freshwater flux $q$ without volume change in the ocean) at the water cell immediately adjacent to the ice interface. In MITgcm, the ice-ocean interface can be horizontal (Losch, 2008) or vertical (Xu et al., 2012). Melting ($q<0$) freshens the ambient water, while freezing ($q>0$) leads to seawater salinification.

Ice-ocean processes are governed by the three-equation model of Hellmer and Olbers (1989) with modifications of Jenkins et al. (2001), which can be written as heat (Eq. A1) and salinity budget (Eq. A2) for the boundary layer, together with the linear



dependency of the freezing point of seawater $T_{\text{fr}}$ with pressure (Eq. A3), in symbols:

$$c_p(\rho\gamma_T - q)(T - T_b) = -Lq - \rho c_{pI}\kappa \frac{T_I - T_b}{\Delta h} \tag{A1}$$

$$(\rho\gamma_S - q)(S - S_b) = -S_b q \tag{A2}$$

$$T_{\text{fr}} = (0.0901 - 0.0575 S_b) - 7.61 \times 10^{-4} p \tag{A3}$$

where $c_p$ = 4000 J kg$^{-1}$ K$^{-1}$ and $c_{p,I}$ = 2000 J kg$^{-1}$ K$^{-1}$ are the specific heat capacity of seawater and ice, $L$=334000 J kg$^{-1}$ is the latent heat of fusion, $\kappa$= 1.54$\times 10^{-6}$ m$^2$ s$^{-1}$ is the heat diffusivity through the ice, $\Delta h$ is the ice shelf draft, $\rho$ is the density of seawater, $T_b$, $S_b$ are temperature and salinity at the boundary layer (water cell adjacent to the ice interface), $T$ and $S$ are local temperature and salinity, $T_I$=-20$^o$C is the core temperature of the ice shelf and $p$ is the local pressure.

The freshwater flux $q$ can be translated into melt rate $\dot{m}$ using the density of ice $\rho_I$ through $\dot{m} = -q/\rho_I$. As we arbitrarily set that $\dot{m} > 0, q < 0$ means melting and $\dot{m} < 0, q > 0$ freezing, larger absolute values of $\dot{m}$ implies stronger melt or freeze rate, depending on the sign. In this study, heat transfer coefficient $\gamma_T$ is held constant to $10^{-4}$, and the salinity transfer coefficient $\gamma_S$ is a function of $\gamma_T$ after the ISOMIP experiments of Losch (2008). By assuming that the temperature at the boundary layer $T_b$ is equal to the freezing point of seawater $T_{\text{fr}}$, equations A1-3 are numerically solved in MITgcm to obtain the triplet $T_b$, $S_b$, $q$ which represents the boundary condition applied at the ice-ocean interface.

**A2 Heat transport toward the ice cavity interior**

The heat transport $HT$ across a meridional cross-section $x = x_1$ can be written as:

$$HT = c_p \iint \rho_\theta(x_1, y, z) u(x_1, y, z) [\theta(x_1, y, z) - T_{fr}] \mathrm{d}y \mathrm{d}z \tag{A4}$$

where $\rho_\theta(x, y, z)$ is the local potential density, $u(x, y, z)$ is the local meridional velocity, $\theta(x, y, z)$ is the local potential temperature and $T_{fr}$ is in-situ freezing point of seawater. Heat-maps of $HT$ integrated across the ice front cross-section $x = 85$ km east for the 120 experiments are shown in Fig. 3. Sensitivity of $HT$ with respect to rift width, boundary current and thermal forcing matches the pattern shown in heat-maps of melt rate in Fig. 2 and is discussed in Sect. 3.1.

**Appendix B: Salt flux across the rift base**

The salt flux $\Phi_S$ across the rift base at depth $z = z_r = 200$ m can be written as:

$$\Phi_S = \frac{1}{A_r} \iint w(x, y, z_r) S(x, y, z_r) \mathrm{d}x \mathrm{d}y \tag{B1}$$

where $A_r$ is the rift base area, $w(x, y, z)$ is the vertical velocity and $S(x,y,z)$ is the in-situ salinity. Heat-maps of $\Phi_S$ integrated across the rift base for the 120 experiments are shown in Fig. B1.



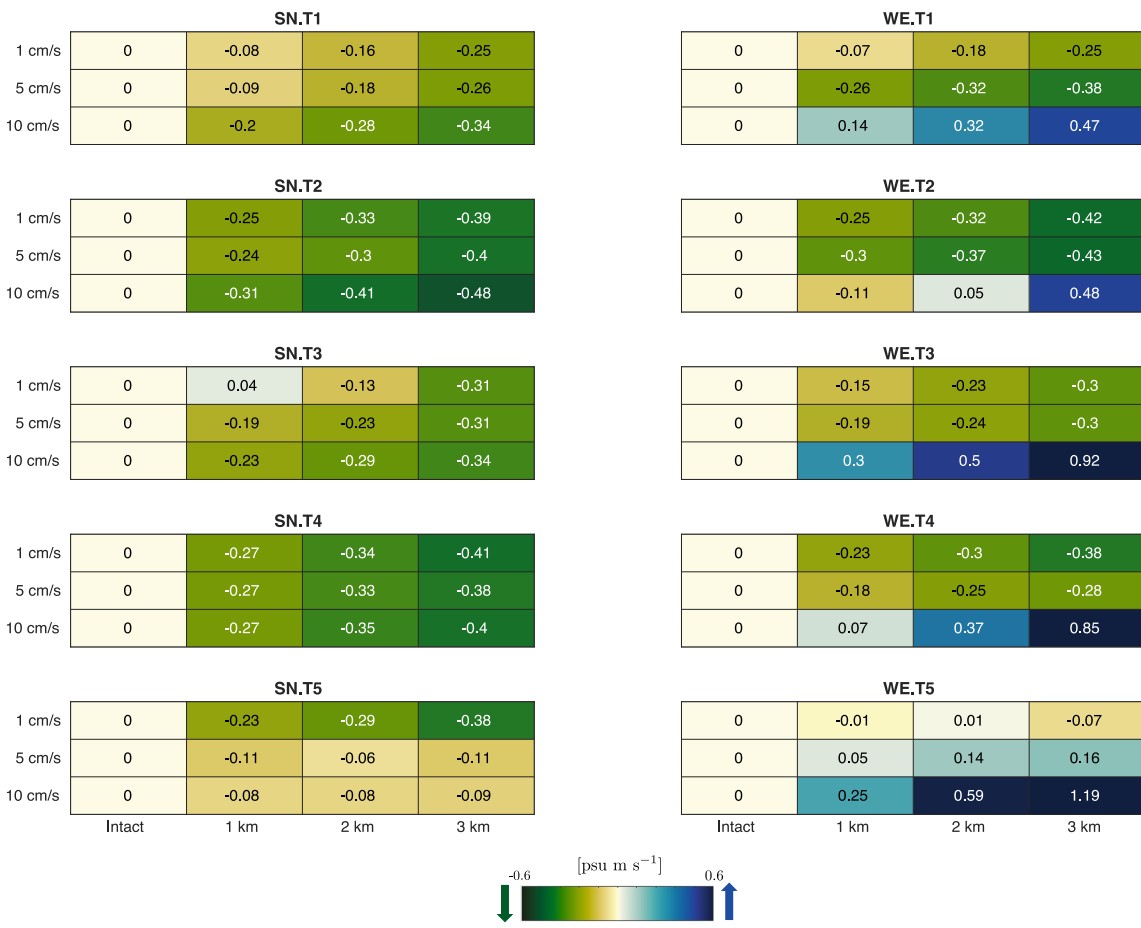

**Figure B1.** Heat-maps of total salt flux $\Phi_S$ across the rift base (200 m) for the 120 sensitivity experiments. Positive and negative values means upward and downward transport, respectively. Values are calculated with respect to temperature profile (panel), flow velocity (row) and rift width (column).

## Appendix C

## Appendix D: Ocean stratification

Ocean stratification can be calculated with the square of the Brunt-Väisälä (buoyancy) frequency, which can be calculated as the vertical buoyancy gradient:

430
$$N^2 = -\frac{g}{\rho_\theta}\left(\frac{\partial \rho_\theta}{\partial z}\right) \tag{D1}$$

where $\rho_\theta$ is the local potential density and $g$ is the gravitational acceleration. $N^2$ is a measure of the stability of a water column to convection/overturning. Positive values of $N^2$ indicate a stable stratified fluid where lighter water overlays denser water. A



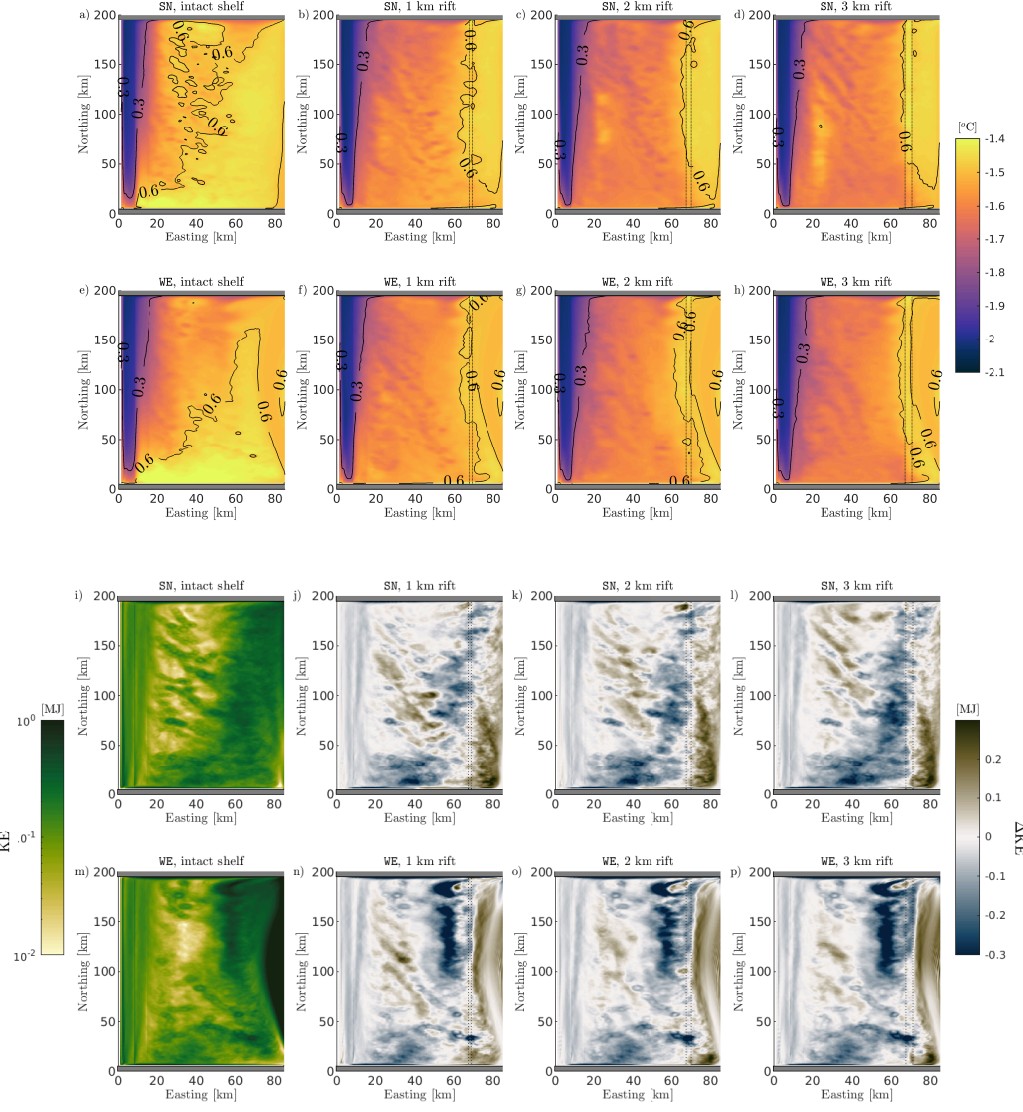

**Figure C1.** Comparison of horizontal distribution of potential temperature and kinetic energy (KE) for different rift widths at the grounding line level (255 m, cross-section $B$ in Fig. 1b). First and third row (second and fourth row) correspond to SN (WE) boundary currents of 5 cm s$^{-1}$, respectively. The cold temperature profile T2 is used for all simulations. Dashed line is the rift location. Panels a-h compare the horizontal distribution of potential temperature (color-map) and the difference between in-situ potential temperature and the freezing point of seawater (black contour lines). Panels i-p compare the kinetic energy. The first column shows the horizontal distribution of kinetic of intact experiments (i,m). Panels j-l and n-p show the horizontal distribution of kinetic energy anomaly, which is calculated as the difference between kinetic energy at 255 m in the rifted shelf simulations with respect to intact cases under the same forcing conditions. Left colorbar refers to kinetic energy, while right colorbar refers to kinetic energy anomaly.





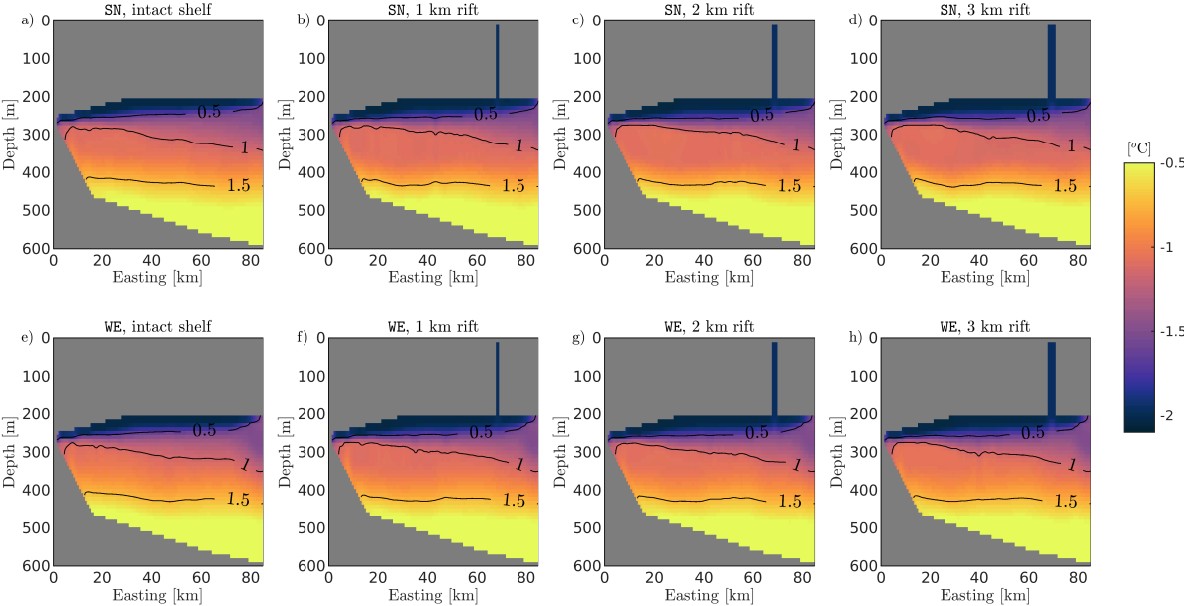

**Figure C2.** Comparison of vertical of potential temperature (color-map) and the difference between in-situ potential temperature and the freezing point of seawater (black contour lines) for experiments with different rift widths at the zonal cross-section along km 50 north. Note that the values in the colorbar values is different than Fig. C1a-h. First and third row correspond to SN (WE) boundary currents of 5 cm s$^{-1}$, respectively. The cold temperature profile T2 is used for all simulations

negative value of $N^2$ indicates an unstable water column where denser water is found above lighter water and convection/over-turning processes increase.

435 *Author contributions.* MP conceived and designed the model simulations, run the experiments, analysed results and wrote the manuscript. MS supported design and model set-up. EL, MV, RR and MS discussed results and revised the manuscript.

*Competing interests.* The authors declare that they have no conflict of interest.

*Acknowledgements.* Part of the research was carried out at the Jet Propulsion Laboratory, California Institute of Technology, under a contract with the National Aeronautics and Space Administration (NASA). We gratefully acknowledge support from the NASA Cryospheric Sciences
440 Program. High-end computing resources were provided by the NASA Advanced Supercomputing Division of the Ames Research Center.





Colormaps used in this manuscript are from Thyng et al. (2016). Finally, MP sincerely thanks Yoshihiro Nakayama, Dimitris Menemenlis, Eric Rignot, Stefanie Nanninga and Ratnakar Gadi for stimulating discussions during the final stage of this research project.



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
