# Peer review of "Can rifts alter ocean dynamics beneath ice shelves?"

_EGUsphere, 2023_

## Author Response (AR1)

RESPONSE TO REFEREE 1 by Mattia Poinelli and co-authors
Referee 1's comments are in italics, while our responses are indented, in regular style text and in red. Additionally, we have provided reference to the specific lines in the track-changes version where the edits have been made.

*General comment*

*The authors present here numerical simulations showing how the presence of an ice shelf rift can act as a barrier to the inflow of off-shelf water, thereby reducing ice melt rates when compared to an ice shelf cavity with no rifting. This is a process which is not currently represented in large scale models and is likely to have implications for accurate predictions of future climate scenarios. Of particular note is that this reduction in melting is found to be concentrated in the proximity of the grounding line, an area of particular sensitivity regarding ice dynamics.*

*I find the manuscript to be generally well written, with an easy-to-follow methodology and results clearly presented. I have no major issues for the authors to address but do have several smaller comments. Assuming the authors address these I recommend publication.*

> We thank the referee for the appreciation and the insightful feedback. We are glad to hear that the message of this paper was received and overall, it was easy to follow.

> Please find below a detailed response to the specific comments and technical corrections that were provided. We have produced a tracked changed version of the manuscript which implements feedback from both referees. Henceforth, we will refer to the old and new version of the manuscript as "previous" and "current", while referring to edits in lines of the "track-changes" version of the manuscript. We agree that these edits will strongly improve clarity as well as the impact of this paper and we thank the referee again for their fundamental input.

*Specific comment*

*In the paragraph beginning L 358 the authors mention some limitations of their modelling framework regarding sea ice. I feel this section could be improved by also including some mention of how the presence of frazil ice formation in supercooled water in the rift would affect their results. For example, the Jordan et al 2014 paper they cite found that the majority of freezing in their simulations was occurring via frazil ice formation rather than directly at the ice-ocean boundary. Would this potentially have the effect of enhancing the barrier to off-shelf water inflow?*

> We agree with the referee that frazil ice accretion in the ice shelf cavity is an important player in the sub-shelf dynamics. As the referee mentions, Jordan et al. 2014 quantifies the extent of frazil ice formation to be > 90% of the total freezing occurring in a rift. At the moment, MITgcm does not include a fully developed "frazil-ice" module, which could account for frazil ice accretion within the water column. Currently, it is possible to check

for the presence of water below the freezing point at depth using the "frazil" package. This set of routines restores the negative heat anomaly of frazil accretion to the surface level within one time step. However, this does not account for the generation or transport of salt anomalies (hence buoyancy gradients) within the water column.

We tend to think that frazil ice formation within the water column - in particular suspended in meltwater plumes in the rift - should substantially affect stratification and buoyancy of the water column with possible repercussion in the "rift barrier" presented here. The extent at which the stratification is actually affected by frazil crystals is however dependent on the size of the crystals themselves (Jordan et al 2014). In particular, smaller crystals may increase the buoyancy while larger crystals may decrease it (Hewitt 2020). As we have not simulation results to support these claims nor have the capabilities/resolution to investigate the extent of our speculations at this time, we mention the absence of frazil ice physics as a limitation of the current study in the methodology section (lines 100-101 in the track-changes version) adding the discussion summarized above in the discussion section (lines 414-426 in the track-changes version).

*When discussing the reduction in melt rates in section 3.2 the results are shown in absolute changes in melt rate for the anomalies plots. I think it might be good to also show the relative change in melt rates as a percentage of the reference case. This would more clearly show if there were a change in the melt pattern or just the magnitude of the melt pattern.*

We thank the referee for this suggestion. We acknowledge the importance of indicating relative changes on top of absolute differences. As a response to the referee's input, we have added Figure C1 with the relative difference between rifted and control run. It is important however to note that there are some areas of the ice shelf that experience a switch from melting to freezing, which in terms of relative value is quantified as more than 100% alteration, even if the absolute change is fairly small.

*I also think that the finding that the biggest reduction in melt rates is seen in the grounding line proximity should be highlighted more in the introduction and conclusions. From an ice dynamic perspective, we know that melt rates near or at the grounding line are the most important, and so highlighting this finding would be of interest to the glaciological community.*

We agree with the referee that melt reduction in proximity of the grounding line represents one of the key points of this manuscript. In response to the referee feedback, we have further highlighted grounding line processes in the introduction (lines 35-37, 47-49, 78-79 in the track-changes version). Furthermore, we have included this specific finding in the conclusion (lines 440-443 in the track-changes version) and added a reference to the anomaly calculated at the grounding line also in the abstract (line 11-12 in the track-changes version).

*When comparing the relative behaviors of the S-N and E-W set ups, would it not be a fairer comparison to have the same total flux of inflow water in each case? Unless I have misunderstood,*

*the S-N case inflow happens over a 15 km section, whilst the W-E happens over a 50 km wide section. It could be argued that the W-E case therefore has a greater supply of water above/below the freezing point before any consideration is made to its direction of flow.*

We thank the referee for this comment. We acknowledge that comparing experiments with the same inflow would be a valid approach and a fairer way to compare SN and WE simulations. However, we chose to fix the velocity instead of water flux due to computational efficiency and the specific goals of our paper.

If we were to fix the water flux instead, we could encounter numerical issues and an increase in computational costs. For example, trying to match the same inflow in both simulations could trigger numerical convergence issues and result in unphysical velocities that could substantially increase the computational effort/duration to spin-up the model. For example, if we were to keep the WE velocity to 10 cm/s ("inflow" of 6Sv), we would need to consider velocities of ~ 0.7 m/s in order to obtain the same transport in SN experiments. This may be a) unphysical given the velocities observed under the ice and b) may also trigger numerical convergence issues (CFL violation). Similar argument can be done if we were to keep the SN velocity to 0.1 cm/s, which would lead to minimum WE velocities of ~ 1 mm/s.

Even though we agree that fixing heat content/total energy in the cavity would be a sensible option for comparing SN and WE experiments, we do not expect it to change the deflection process along the rift that is the main focus of our paper. We do realize that we still compare SN and WE simulations in various results sections, but only as a means of presenting numerical outcomes, and not as a main objective.

*I would also find it more intuitive to have the ice shelf orientated North-South rather than East-West for the descriptions of scenario labels, though I admit this is a personal preference and not a requirement by any means.*

We thank the referee for the perspective on the model orientation. Our view on the geometry is referring to a standard f-plane approximation, where the Cartesian (orthogonal) coordinates x-y are oriented as longitude-latitude. Our approach to describing SN cases first is to "fix" the flow in longitude (x), before describing WE cases where the flow is "fixed" in latitude (y). We also agree that this is purely based on personal preference.

*Technical corrections:*
*L 34 - "Albeit critical, ...." Reword for clarity, as it currently reads as if the pathways themselves are elusive rather than the observations of them.*
*L 55 - First use of Tfr, need to define.*
*L 56 - atmosphere -> atmospheric*
*L 57 - relative _> relatively*
*L 64 Would read clearer as "water formed within them"*

*L 93 I assume intensity is referring to inflow velocity?*
*L 95 "as a discontinuity"*
*L 98 " Despite the fact ice "*
*L169 "that computed".*
*L169 "under an identical"*
*L198 "a negligible"*
*L 205 "to a progressively "*
*L 234 "the inland-most"*
*L 237 "freezing within".*
*L 253 "being the same".*
*L 318 "tends to flow".*

We thank the referee for their time providing us with technical corrections and hints on typos. In the current version of the manuscript, all these corrections have been addressed.

*General comments*

*This highly idealised model study investigates the effects of a rift across a floating ice tongue in an ice shelf, using the MITgcm numerical model. As a result, import of relatively warm off-shelf water into the ice cavity is decreased by the presence of a rift and further decreases with increasing width of the rift.*

*Generally, the manuscript is well-written and the results are convincing. There are many grammatical errors of which I have just noted a few. The design of the idealised model setup needs more motivation. The ice-ocean interface is quite flat (just extending from 250m at the grounding line to 200 m at the calving front), and there is no sill. Some discussion of these simplifications and their impact on the results is needed. The authors criticise existing more realistic and larger-scale models that they do not resolve such rifts. Those rifts are however, as mentioned in the discussion highly intermittent at short time scales of days, and it is unclear how this relates to the simulations performed here that are run for 7 years with a stationary under-ice topography. It would be good if the authors could propose a concept how these subgridscale processes can be considered in numerical models in general.*

*More discussion and explanation is needed in general. These suggested modifications of the manuscript are largely minor.*

> We appreciate the detailed and insightful feedback from the referee. In response to the general and specific comments provided, we have included further discussion and explanation both in the methodology and discussion section. In order to address motivations and explanations inquired by both referees, we substantially re-structured section 2.1: "Model set-up", which now integrates a large part of section 2.2 "Simulations design".

> First of all, we added more motivation to the idealized model set-up (flat and shallow ice draft, bathymetry and absence of a bathymetric sill) in section 2.1. We address these motivation later in this response. Our goal was to idealize the geometry as far as possible and, in our opinion, the chosen geometry takes into account the most prominent features driving the circulation under Antarctic ice shelves. Secondly, it is true that rifts on large Antarctic ice shelves can propagate at nearly instantaneous rates, similar to fractures at the glacier terminus in Greenland. However, it is also important to note that rifts on Antarctic ice shelves can remain dormant for extended periods, ranging from months to years. For example, the A-68 rift on the Larsen C ice shelf was first detected in 2005 and slowly propagated for less than 100km over the course of approximately 10 years

(Borstad, 2017). Additionally, multiple rift features were observed near the ice front of the Ronne ice shelf in 1987 (Swithinbank, 1987), which became more apparent in 1994 (King, 1994), i.e. 7 years after. Similarly, the "lose-tooth" rift system in the Amery ice shelf remained active over several years with apparent propagation rates of 2-5 m/day (i.e., a few km per year) (Fricker, 2005). We acknowledge that without mentioning this "long term " processes, the reference to rifts may be unclear and lead to questions regarding their fast evolution. As a response to the referee's comment, we have included a brief discussion on the duration of rifts in the introduction (lines 56-60 in the trak-changes version). The main focus of our work is to investigate the presence of these "long-term" rifts. Lastly, we have included limitations and caveats of this model due to the resolution employed here which does not resolve sub-grid scale processes mentioned by the referee.

Please find a detailed response to the specific comments below this point. We have produced a tracked changed version of the manuscript which integrates feedback from both referees. Henceforth, we will refer to the old and new version of the manuscript as "previous" and "current", while referring to edits in lines of the "track-changes" version of the manuscript. We agree that these edits will strongly improve clarity as well as impact of this paper and we thank the referee again for their fundamental input.

*Specific comments*

*16: How can you have an acceleration of 94 Gt/yr each decade at a current rate of 134 Gt/yr? Does it mean that the rate was negative until two decades ago?*

We thank the reviewer for finding this mistake. By using revised inventories, improved thickness mapping, and time series of velocity and SMB, Rignot et al. 2019 estimated an acceleration of 94 Gt/yr per decade between the investigated timeframe 1979–2017, increasing from an average rate of 48 Gt/y per decade in 1979–2001 to 134 Gt/y per years between 2009–2017. In the previous version of the manuscript, we mistakenly wrote that the ablation rate between 2001 and 2017 was 134 Gt/yr. We decide to only mention the mean ablation rate between 2009 and 2017, which is 252 Gt/yr.

*49: A brief further explanation of the "compressive arch" might be helpful.*

In response the referee's comment, we have added a brief explanation of the "compressive arch" (lines 53-56 in the track-changes version).

*53: typo „orders or magnitude".*

We thank the referee for pointing out this typo.

*63: „limited to a 2-D rift environment": which two dimensions are meant here? Two horizontal or one vertical and one horizontal?*

Jordan 2014 and Khazendar 2003 developed 2D model set-ups with "vertical" and "horizontal' degrees of freedom. After the referee's input, we have add vertical-horizontal between parenthesis (line 74 in the track-changes version).

*78: How much can potentially important non-hydrostatic effects at vertical side walls of 200 m thickness be ignored in a 1 km wide ice rift? Did you use any of the entrainment formulations existing for vertical melt water plumes?*

As the referee points out later, a resolution of 10m does not resolve vertical melt-water plumes (Xu 2012 and Burchard 2022). Furthermore, we employ a fully hydrostatic formulation. Therefore, non-hydrostatic effects such as melt-driven plumes and the physics of their entrainment are not considered here. For example, the model is not capable to distinguish whether the plume reaches the natural buoyancy and detaches from the rift wall (e.g. Khazendar 2003). Moreover, frazil accretion processes when the water reaches the supercooled state within the water column are also neglected. We do realize that these are important caveats of our idealized model and, in response to the referee's comment, we further specify the limitations of our approach in the methodology (lines 100-101 in the track-changes version) and discussion section (lines 414-426 in the track-changes version).

*86: Please indicate which latitude this Coriolis acceleration corresponds to.*

It corresponds to a latitude of 75S. We include this information in the current version of the manuscript (lines 97-98 in the track-changes version).

*87: With a vertical resolution of 10 m, subglacial plume are not vertically resolved, see the discussion by Burchard et al. (2022). What consequences does this have for the vertical heat transport towards the ice-ocean interface?*

This is correct. The resolution employed here does not resolve melt-driven plumes (also discussed in Xu 2012). This is an important caveat of our approach which is better explained in the new version of the manuscript. We tend to think that one of the most important consequence of neglecting plumes - beside missing the physics of the entrainment and heat transports between ambient water and plume - comes with the omission of frazil accretion processes, which rift are filled with (Orheim1990, Lawrence 2023). The size of these frazil crystals would impact the buoyancy of melt-driven plumes. In particular, smaller crystals may increase the buoyancy while larger crystals may decrease it (Hewitt 2020). However, we do not have numerical simulations nor the capability to support these claims. We do recognize these important caveats in the design of our idealized model and, in response to the referee's comment, we address the limitations of our approach in the methodology (lines 100-101 in the track-changes version) and discussion section (lines 414-426 in the track-changes version).

*91: Please indicate for which ice shelves a grounding line depth of 250 m is characteristic. To my knowledge the large ice shelves have much deeper grounding lines. Also, no sill has been imposed at the bottom. Which impact does this simplification have compared to typical ice shelf dimensions?*

We do realize that a grounding line depth of 250 m may be relatively shallow and that the depth at which ice detach from the bedrock is critical for ablation processes, especially because of the pressure dependance of the freezing point of seawater. However, areas of some large (~100-200 km) ice shelves in Antarctica are characterized by relatively shallow grounding zones. E.g. southern areas of the Larsen C ice shelf show grounding lines at depths between 200 - 300 m with a relatively flat ice base geometry. In the WAIS, notorious examples are the Wilkins and Bach Ice shelves with a grounding line depth at approximately 250 m. In the Queen Maud land, examples are the Roi-Baudouin and the Borchrevink ice shelves. Similarly, there are ice shelves where sills are still not a common characteristic (e.g. Larsen C). Despite bathymetric sills and ridges impact the sub-shelf circulation by squeezing the water column and therefore act as topographic barrier, we do not consider their impact here. In response to the referee's comment, we have included a reference to these ice shelves as well as the motivation for the chosen geometry in section 2.1 (lines 121-129 in the track-changes version).

We do realize that a highly-idealized model like the one we presented may have substantial biases in the outcome of sub-shelf circulation patterns based on the assumptions we made. For example, we also fix the bathymetry as a prograde slope, neglecting important implications of ice shelves laying on retrograde bed. However, given our goal to estimate the impact of rifts in the heat transport toward the ice shelf cavity using multiple rift dimensions, off-cavity flows characteristics, we idealized our geometry as far as possible.

*98: Better „sea ice".*

We thank the referee for pointing out this typo.

*Section 2.2: Please mention that no subglacial discharge was prescribed at the grounding line and discuss the implications. Do also mention that ice melt does not move the ice-ocean interface.*

We thank the referee for the suggestion. In the current version of the manuscript, we mention the absence of subglacial discharge (lines 127-129 in the track-changes version )and the fixed ice shelf geometry (lines 111-112 in the track-changes version) as limitations of our model set-up. We further discuss the omission of subglacial ice discharge in the discussion section (lines 395-400 in the track-changes version).

*140-141: Isn't cross-section B showing the „horizontal distribution of tr01 across a horizontal cross-section"?*

That is correct, cross-section B is horizontal. We thank the referee for pointing out this mistake.

*144: „heat fluxes (in the form of melt rate)", not clear what is meant here. Heat flux and melt rate have a complex non-linear relation, how can heat fluxes then be presented in the form of a melt rate?*

We agree that simply referring to "heat fluxes" without specifying the context can be misleading and confusing. We referred to heat flux in the context of the 3-equation-model (equations are sketched in Appendix A1), where the freshwater flux is calculated as the heat flux across the ice-ocean interface divided by the latent heat of fusion (eq. A1-3 in Losch 2008). In response to the referee's comment, we have removed "heat fluxes" in the current version of the manuscript and replace it with "melt rate", leaving the definition of heat and freshwater fluxes in Appendix A1.

*185-186: typo „in almost all experiment".*
*222: should be „re-freezes".*
*224: Shouldn't it be „upwelling"?*

We thank the referee for pointing out these typos.

*229-230: What do you mean with „stronger off-shelf water"?*

We replace "stronger" with "faster".

*234: Do you mean „reach the inland-most section"?*

Yes, we thank the referee for pointing out this typo.

*235-237: This looks like an artefact of the experiment. To widen the rift, also the west flank could be moved further west. By widening the east flank towards the calving front, two effects are mixed up: closer proximity to the open ocean and a wider rift.*

We thank the referee for this observation. That is right, it may be a consequence of the model set-up. The rift can be "widened" also by moving the west flank westward. By moving the rift wall to the east, the interaction with the open ocean become more intense as it can be seen in figure 6l at depth of ~200 m, where higher melting occurs after the deeper sections of the rift are in contact with water. In response to the referee's comment, we have mentioned the potential limitation of our claims (lines 273-275 in the track-changes version).

*Fig. 7, panels i-p: Since only one contour line is shown, it cannot be seen where minima and maxima occur. It would be better to show a different contour line in a different colour.*

We thank the referee for the suggestion. Figure 7i-p and Figure 9i-p now include a second contour line (0.03 in magenta) to better visualize concentration gradients.

*262: What is the thickness of this buoyant plume as calculated by the KPP model and how is it resolved by the vertical discretisation?*

As pointed out by the referee, a vertical resolution of 10 m and the employed hydrostatic formulation do not resolve melt-driven plumes nor processes controlling their entrainment (Burchard 2022, Xu 2012). Rather than a "buoyant plume", the model simulates a layer of cold water (-1.9 C) that is adherent to the ice base. In response to the referee comments, we realize that labeling "plume" may be confusing given the limitations of our model. As so, we have removed the word "plume" which is inaccurately used in line 262 of the previous version of the manuscript and we replace it with "buoyant meltwater".

*268: Also for the case without rift (Fig. 7m), the concentration of tracer 1 is very low near the calving front.*

That is correct, we thank the referee for pointing out the possible confusion in this sentence. In the intact case, low values of tr01 (we meant on the order of 0-0.1) along the southern bound are simulated all the way to km 20 East while in the rifted case concentrations of 0-0.1 are confined seaward of the rift. We have corrected the paragraph accordingly (lines 303-304 in the track-changes version).

*290-291: This indicates a 75 m thick buoyant plume. Is that what you expect under Antarctic ice shelves or is this a model artefact?*

Rather than being solely influenced by the presence of buoyant plumes, the Brunt-Vaisala frequency indicates the stratification of the water column based on its density gradient (eq. D1). Cold water ice shelves can have higher stratified layers that range from 20 m to 70 m in thickness (e.g. figure 2b,c in Davis 2019). We agree that the presence of buoyant plumes can complicate these processes by altering the stratification of the mixed layer that adheres to the ice base (Vreugdenhil 2019). However, non-hydrostatic processes are not explicitly represented here due to the limited resolutions used in our model and similarly our idealized set-up does not resolve melt-drive plumes nor processes driving their entrainment.

*304: should be „leads to spreading of saltier and denser water".*

We thank the referee for pointing out this typo.

*306-307: Isn't this mainly because the initial tracer contents in wider rifts is much larger than in narrower rifts?*

That is correct, larger rifts are containing a larger concentration with respect to narrower rifts. As a response to the referee's comment, we have mention this issue in the current version of the manuscript (lines 342-343 in the track-changes version).

*318: should be „tends"*

We thank the referee for pointing out this typo.

*347-350: This shows the potential impact of subglacial discharge that has been neglected here. A doiscussion of the effects of this ommission is needed here.*

We thank the referee for this suggestion. As mentioned earlier, we have included a clarification in section 2.1 specifying that our idealized model does not account for sub-glacial discharge at the grounding line. We have also added a short paragraph in the discussion section (lines 395-400 in the track-changes version) describing the implication of freshwater discharge in past numerical efforts.

*354: typo „As an precise representation".*

We thank the referee for pointing out this typo.

References:

Burchard, H. et al. (2022). The vertical structure and entrainment of subglacial melt water plumes. Journal of Advances in Modeling Earth Systems, 14(3), e2021MS002925.

Borstad, C., Rignot, E., Mouginot, J., & Schodlok, M. (2013). Creep deformation and buttressing capacity of damaged ice shelves: theory and application to larsen c ice shelf. Cryosphere, 7.

King, E. (1994). Observations of a rift in the Ronne Ice Shelf, Antarctica, Journal of Glaiology (134), 187-189. doi:10.3189/S0022143000003968.

Swithinbank, C., Brunk, K. and Sievers, J. 1987. Glaciological map of Filchner-Ronne-Schelfeis . Frankfurt am Main, Institut für Angewandte Geodäsie.

Fricker, H. A., N. W. Young, R. Coleman, J. N. Bassis, and J.-B. Minster (2005), Multi-year monitoring of rift propagation on Amery Ice Shelf, East Antarctica, Geophysical Research Letters, 32(2), L02502, doi:10.1029/2004GL021036.

Rignot, E. et al. .: Four decades of Antarctic Ice Sheet mass balance from 1979–2017, Proceedings of the National Academy of Sciences, 116, 1095–1103, https://doi.org/10.1073/pnas.1812883116,605 2019

Jordan, J. J. et al.: Modeling ice-ocean interaction in ice-shelf crevasses, J. Geophys. Res.: Oceans, 119, 995–1008, https://doi.org/doi.org/10.1002/2013JC009208, 2014.

Khazendar, A. and Jenkins, A.: A model of marine ice formation within Antarctic ice shelf rifts, J. Geophys. Res., 108, https://doi.org/10.1029/2002JC001673, 2003

Orheim, O. et al.: Glaciological and oceanographic studies on Fimbulisen during NARE 1989/90,585 in Filchner-Ronne Ice Shelf Programme, edited by H. Oerter, Rep. 4., 120–129, 1990

Lawrence, J. D. et al.: Crevasse refreezing and signatures of retreat observed at Kamb Ice Stream grounding zone, Nature Geoscience, 16, 238–243, https://doi.org/10.1038/s41561-023-01129-y, 2023.

Hewitt, I. J.: Subglacial Plumes., Ann. Rew. of Fl. Mech., 52, 145–169, 2020.

Losch, M.: Modeling ice shelf cavities in a z coordinate ocean general circulation model, J. Geophys. Res.: Oceans, 113, https://doi.org/10.1029/2007JC004368, 2008.

Xu, Y. et al.: Numerical experiments on subaqueous melting of Greenland tidewater glaciers in response to ocean warming and enhanced subglacial discharge., Ann. Glaciol., 53, 229 – 234, https://doi.org/10.3189/2012AoG60A139, 2012

Davis, P. E. D., & Nicholls, K. W. (2019). Turbulence observations beneath larsen c ice shelf, antarctica. J. Geophys. Res.: Oceans, 124

Vreugdenhil, C. A. and Taylor, J. R.: Stratification Effects in the Turbulent Boundary Layer beneath a Melting Ice Shelf: Insights from Resolved Large-Eddy Simulations., Journal of Physical Oceanography, p. 1905–1925, https://doi.org/10.1175/JPO-D-18-0252.1, 2019